# PathfinderTURB: an automatic boundary layer algorithm. Development, validation and application to study the impact on in-situ measurements at the Jungfraujoch.

Yann Poltera[1,3], Giovanni Martucci[1], Martine Collaud Coen[1], Maxime Hervo[1], Lukas Emmenegger[2], Stephan Henne[2], Dominik Brunner[2] and Alexander Haefele[1]

[1] Federal Office of Meteorology and Climatology, MeteoSwiss, Payerne, Switzerland

[2] Swiss Federal Laboratories for Materials Science and Technology, Dübendorf, Switzerland

[3] Institute for Atmospheric and Climate Science, ETH Zurich, Zurich, Switzerland

**Abstract.** We present the development of the PathfinderTURB algorithm for the analysis of ceilometer backscatter data and the real-time detection of the vertical structure of the planetary boundary layer. Two aerosol layer heights are retrieved by PathfinderTURB: the Convective Boundary Layer (CBL) and the Continuous Aerosol Layer (CAL). PathfinderTURB combines the strengths of gradient- and variance-based methods and addresses the layer attribution problem by adopting a geodesic approach. The algorithm has been applied to one year of data measured by two ceilometers of type CHM15k, one operated at the Aerological Observatory of Payerne (491 m, a.s.l.) on the Swiss plateau, and one at the Kleine Scheidegg (2061 m, a.s.l.) in the Swiss Alps. The retrieval of the CBL has been validated at Payerne using two reference methods: (1) manual detections of the CBL height performed by human experts using the ceilometer backscatter data; (2) values of CBL heights calculated using the Richardson's method from co-located radio sounding data. We found average biases as small as 27 m (53 m) with respect to reference method 1 (method 2). Based on the excellent agreement with the two reference methods, PathfinderTURB has been applied to the ceilometer data at the mountainous site of the Kleine Scheidegg for the period September 2014 till November 2015. At this site, the CHM15k is operated in a tilted configuration at 71° zenith angle to probe the atmosphere next to the Sphinx Observatory (3580 m, a.s.l.) on the Jungfraujoch (JFJ). The analysis of the retrieved layers led to the following results: the CAL reaches the JFJ during 41% of the time in summer and during 21% of the time in winter for a total of 97 days during the two seasons. The season-averaged daily cycles show that the CBL height reaches the JFJ only during short periods (4% of the time), but on 20 individual days in summer and never during winter. Especially during summer the CBL and the CAL modify the air sampled in-situ at JFJ, resulting in an unequivocal dependence of the measured absorption coefficient on the height of both layers. This highlights the relevance of retrieving the height of CAL and CBL automatically at the JFJ.

## 1 Introduction

During convective periods, particles and gases are mixed homogeneously within the convective boundary layer (CBL). The upper limit of the CBL corresponds to the interface between the well-mixed region and the free troposphere (FT) above it. This interface, also called entrainment zone (EZ), is a turbulent transition of few tens to few hundreds of meters thick characterized by negative buoyancy flux. The study of the EZ and the way the CBL air is mixed through it has drawn particular attention during the last decades. There are various methods to study the CBL and the EZ, based on profiles of temperature, backscatter or turbulence measured either by radio-

sounding or by passive and active remote sensing or calculated by numerical models. Amongst the different observational methods, the remote sensing technique ensures the largest amount of profile data. Active remote sensing (acoustic or laser-based) provides the best vertical resolution allowing to resolve the multiple transitions (including the EZ) between different layers in the CBL and the FT. Probably the best-suited instrument to study these dynamics at high temporal and vertical resolution is the ceilometer, a low-power, compact and cost-effective version of a research LIDAR. A ceilometer is a laser-based instrument normally emitting in the near-infrared spectral band (800-1100 nm), highly sensitive to aerosols and cloud droplets. In the early 2000's, the first manufacturers (e.g., Vaisala, Leosphere, MPL, Jenoptik) started producing ceilometers with the capability to store the entire backscatter profile in addition to the cloud base height. Rapidly, ceilometers have been recognized by the meteorological services and research centres in Europe and worldwide as an efficient and affordable way to study the troposphere using aerosols as tracers (e.g., Münkel, 2007; Flentje et al., 2010; Martucci et al., 2010; Heese et al., 2010; Wiegner and Geiss, 2012; Wiegner et al., 2014). Over the last decade, ceilometers have increased significantly in number especially in Europe, United States and Asia , now reaching nearly 1000 units only in Europe (http://www.dwd.de/ceilomap). If combined in a single large network, all ceilometers could provide helpful information on the vertical and horizontal distribution of aerosols and on the status of the CBL over a very large geographical domain in near real time.

In order to process automatically a large amount of data over a large and geographically-diverse domain, we need an algorithm capable to retrieve the vertical structure of the boundary layer (BL) during both convective and stable conditions and over both flat and complex terrain. The conditions inside the stable BL (SBL) are generally stratified and characterized by strong radiative cooling especially during clear nights. On the other hand, the CBL is characterized by an active mixing due to the daytime cycle of thermals updraft and downdraft. Several aerosol layers can form inside the BL (and into the FT by advection), so the difficulty of discriminating one layer from another is directly proportional to the number of layers. An efficient retrieval method shall solve the *attribution problem* (layer categorization), i.e. shall detect unambiguously the different aerosol layers and the EZ. The attribution problem still remains one of the major sources of uncertainty related to the CBL and SBL height retrieval. In order to address the attribution problem, we have further developed the *pathfinder* algorithm originally described by de Bruine et al. (2016). We have then validated our own-developed version of pathfinder algorithm and applied to real-time detections of the vertical structure of the BL above complex terrain. This improved version of the pathfinder algorithm, is called PathfinderTURB (pathfinder algorithm based on TURBulence), to highlight the use of aerosol distribution temporal variability (variance) to detect the BL height. PathfinderTURB has been applied to the data of a ceilometer installed at the Kleine Scheidegg to probe the air sampled by the in-situ instrumentation at the high Alpine station Jungfraujoch (JFJ). The JFJ is part of numerous global observation programs like GAW (Global Atmospheric Watch), EMEP (European Monitoring and Evaluation Programme), NDACC (Network for the Detection of Atmospheric Composition Change) and AGAGE (Advanced Global Atmospheric Gases Experiment). Most importantly, in the context of this study, JFJ participates with in-situ observations as a level-1 station in the ICOS project. In contrast to other ICOS sites located over flat terrain, it was decided to install the ceilometer at KSE to characterize the CBL below and above the JFJ. The presence of the aerosols detected by the ceilometer and the frequency at which these reach the JFJ, are directly compared to the optical, chemical and physical in-situ measurements of aerosols and trace gases at

the JFJ. Several in-situ instruments are installed at the JFJ and operate continuously since many years to measure aerosols, trace gases and several meteorological parameters (Bukowiecki et al., 2016). Instruments of direct interest to our study are: a condensation particle counter (CPC; TSI Inc., Model 3772), which measures the particle number concentration and two instruments providing aerosol absorption coefficients: a Multi-Angle Absorption Photometer (MAAP) measuring at 637 nm and an aethalometer (AE-31, Magee Scientific) measuring at seven different wavelengths. The fact of measuring remotely with a ceilometer the presence of the CBL air in real time and close to the JFJ for more than one year is unprecedented. A recent study by Zieger et al (2012) has used a scanning LIDAR tilted at 60° Zenith angle for 9 days to probe the air close to the JFJ . Also based on the results of the study by Zieger, we have decided to improve their instrument set-up and to install a ceilometer probing even closer (few meters) to the JFJ and for more than one year. This has allowed to create a statistics of CBL-events and to describe quantitatively the relation between the CBL dynamics (rising and falling) and the aerosols optical properties at JFJ. The relevance of such measurements becomes also clear in the framework of ICOS, where the detection of the CBL height in the vicinity of a level-1 ICOS stations is a requirement to validate the atmospheric transport models. This is crucial when observations of atmospheric compounds at different concentrations must be translated into greenhouse gas fluxes between the atmosphere and the land surface.

## 2    Overview of existing algorithms

Traditionally, the retrieval of the BL height from the backscatter profile of a LIDAR can be done using two types of methods: (i) the gradient-based algorithms that track gradients in the vertical distribution of aerosols (gradient of the backscatter profiles), (ii) and the variance-based algorithms that track fluctuations in the temporal distribution of aerosols (variance of the backscatter profiles). Some algorithms combine both techniques, which makes the BL height retrieval more robust especially in convective conditions when the BL dynamics change rapidly.

The gradient-based algorithms retrieve the BL height by tracking the well-marked drop in the aerosol concentration that often occurs at the base of (or within) the EZ in convective conditions or at the level of the temperature inversion capping the residual layer (RL), in neutrally-stratified conditions. All vertical negative gradients found starting from the ground are transitions between different aerosol layers and correspond to peaks along the LIDAR backscatter gradient profile. All peaks are labelled as possible candidates of the BL height (*layer attribution*) at each time step. The traditional approach using numerical approximations of the first or second derivatives of the LIDAR signal (e.g., Menut et al., 1999), has been improved by using the wavelet covariance transform and the fact that the strong gradient occurring at the top of a layer exists at both small and large scales allowing the wavelet-based methods to reduce the uncertainty when assigning the BL height (Davis et al., 2000; Cohn and Angevine, 2000; Brooks, 2003; Baars et al., 2008; Angelini et al., 2009; de Haij et al., 2010; Frey et al., 2010). Alternatively, the Derivative of Gaussian wavelet is used in Morille et al. (2007) or the Daubechies wavelets in Engelbart et al. (2008). The Canny edge detection method (Canny, 1986) also helps improving the retrieval of aerosol layers (e.g. STRAT2D: Morille and Haeffelin, 2010). It is also worth to mention the method proposed in Steyn et al. (1999), which consists of fitting an idealized backscatter profile at the transition between the BL and the FT. In the more recent literature there are examples of different methods combining the LIDAR gradient-based retrievals with temporal height-tracking techniques, for example

observational (Martucci et al., 2010a, b), predictive (Tomás et al., 2010) or model-based first guesses (Giuseppe et al., 2012). Pal et al. (2013), proposed a simplified bulk-model combined with surface turbulence measurements and atmospheric variance measurements, to help selecting the BL height amongst all candidates. Collaud Coen et al., (2014), used a gradient-based temporal continuity criterion to reduce the problem's degeneration and improve the attribution skill. In the study described by de Bruine et al. (2016) presenting the *pathfinder* algorithm, the gradient field and guiding restrictions are taken as core information to retrieve the BL height based on the identification of the most cost-effective path (called a geodesic) along the gradient lines in a graph.

The variance-based algorithms use the temporal fluctuations in the aerosol backscatter as a function of the height $z$ to retrieve the BL height. Within the EZ, cleaner, drier free tropospheric air is entrained repeatedly and mixed-in with the rising aerosol-laden, moister air coming from the BL. A variance-based algorithm can detect the BL height at the level where the backscatter variability reaches a maximum at the base or within the EZ. Variance-based algorithms calculate the temporal variance of the backscatter profile at each range bin, usually over periods shorter than one hour. Similarly to the gradient-based, the variance-based algorithms use peaks in the smoothed variance profile as candidates for the BL height (e.g., Hooper and Eloranta, 1986; Piironen and Eloranta, 1995; Menut et al., 1999; Martucci et al., 2007).

Because the transitions between different aerosol layers and between the BL and the FT are characterized by both a sharp gradient in aerosol concentration and by mixing of air through the interface, the variance- and gradient-based algorithms normally provide similar retrievals of the BL height. Still, the gradient-based and the variance-based algorithms have their specific advantages and disadvantages under different atmospheric conditions. Indeed, the depth and structure of the BL depend on non-linear interactions at different timescales, induced by mechanical and thermodynamic mixing. When retrieving the BL height it is then important to include in the algorithm more than one source of information (e.g. gradient, variance, a priori information) in order to account for the largest number of atmospheric conditions and then to minimize the attribution uncertainty. Combining the variance- and gradient-based methods allows to compare the two retrievals at each time step (Lammert and Bösenberg, 2006; Martucci et al., 2010a,b, Toledo et al., 2014). The retrieval method STRAT+ (Pal et al., 2013), based on STRAT2D (Morille and Haeffelin, 2010), uses the Canny edge detection applied to gradient profiles along with the information brought by the variance profiles and by the radiosoundings to detect the main BL height and internal boundaries as well as the growth rate.

## 3    Description of instruments and sites

Two ceilometers of type CHM15k-Nimbus (hereafter referred to as only CHM15k) manufactured by Lufft have been deployed for this study at two sites in Switzerland: the Aerological observatory of MeteoSwiss at Payerne (PAY, 491 m a.s.l., 46.799° N, 6.932° E) and the Kleine Scheidegg (KSE, 2061 m a.s.l., 46.547° N, 7.985° E). The CHM15k is a bistatic LIDAR with a Nd:YAG solid-state laser emitting linearly-polarized light at a wavelength of 1064 nm. It has a repetition rate ranging between 5 and 7 kHz, a maximum vertical resolution of 5 m, a maximum range of 15 km, a first overlap point at 80 m and a full overlap reached at 800 m (specific for KSE and PAY ceilometers, Hervo et al., 2016). The standard instrument output is the background-, range- and overlap-corrected, normalized signal $S$ defined at range $r$ and time $t$ as:

$$S(r,t) = \frac{(P(r,t) - B(t))r^2}{C_{CHM}(t)O_{CHM}(r)} \tag{1}$$

Where, $B$ is the background, $C_{CHM}$ is a normalization factor accounting for variations in the sensitivity of the receiver and $O_{CHM}$ is the temporally-constant overlap function provided by the manufacturer. At both sites, PAY and KSE, the overlap function $O_{CHM}$ has been corrected for temperature variations following Hervo et al. (2016).

## 3.1 Sites description

The PAY site is situated in the centre of the Swiss Plateau between the Jura Mountain range (25 km to the northwest) and the Alpine foothills (20 km to the southeast), as it is shown in the upper panel of Figure 1. The measurement site is characterized by a rural environment leading to biogenic aerosols sources combined with moderate urban emissions characterized by anthropogenic aerosol sources especially related to cars exhaust and house heating. PAY is equipped with numerous meteorological measurements allowing to interpret and to validate the measurements from the CHM15k. The most relevant measurements and instruments in the framework of the presented study are the operational Meteolabor SRS-C34 radiosondes launched twice daily at 00 and 12 UTC (Philipona et al, 2013), and the surface sensors of temperature and humidity. The measurements used for this study at PAY have been collected during the period January-December 2014.

The KSE is located in the Bernese Oberland Alpine region, (Figure 1, bottom panel). KSE is on a saddle point between the mountain peak Lauberhorn (2472 m a.s.l.) to the northwest and the Jungfraujoch (3465 m a.s.l.) to the southeast, and it is a pass between the semi-urban areas of Wengen and Grindelwald. This topographic configuration has a considerable influence on the local wind circulation. Winds at the KSE are mostly blowing along the SW-NE axis (Ketterer et al., 2014), whereas the prevailing wind at JFJ are from NW toward SE. The JFJ itself is located on the ridge formed between the Mönch and the Jungfrau mountains and is 4.5 km to the south-east and 1.5 km higher than KSE. Most of the atmospheric observations at the JFJ are obtained at the Sphinx observatory (3580 m a.s.l.).

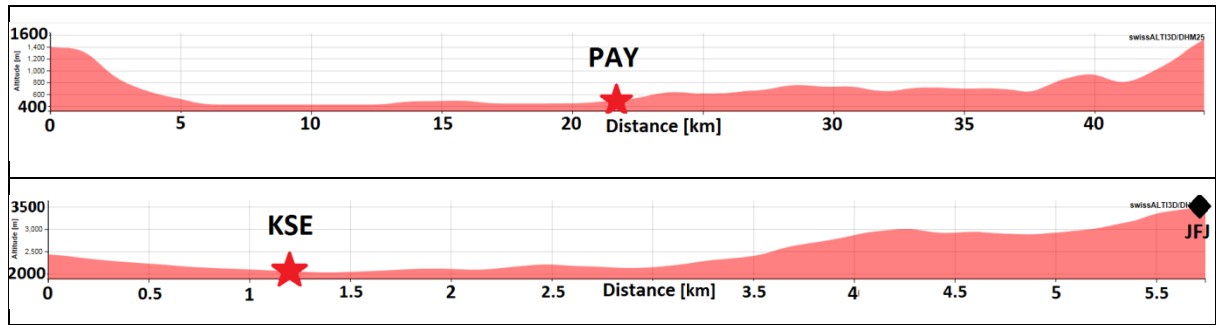

Figure 1. topography of PAY (elevation profile along the 127.2° Azimuth), and KSE (elevation profile along the 151.6° Azimuth) as provided by the federal office of topography (http://www.geo.admin.ch/). The red stars mark the position of the ceilometers at PAY and KSE, the black diamond marks the JFJ position.

## 3.2    Special instrument settings for KSE

The CHM15k ceilometer at KSE was installed in August 2014 on the roof of the maintenance centre of the train station. From September to November 2014 and from March to November 2015, the ceilometer was tilted at 71° zenith angle with the laser beam passing close above (~20 m) the JFJ. From the beginning of November 2014 till the end of February 2015, the ceilometer was set back to the vertical position (5° zenith angle) to prevent the sun shining directly into the ceilometer's telescope.

The tilted setup of the ceilometer was chosen to observe the injections of CBL air at the level of JFJ and to probe the same air as it is measured by the in-situ instruments at the JFJ. When measuring in slant path the maximum vertical height, $R_{max}$, depends on the tilt angle and on the instrument's maximum range (15 km for the CHM15k), at 19° elevation angle $R_{\max} = 2.069 + 15\sin(19°) = 6.64$ km a.s.l.. This value of $R_{max}$ corresponds to a level in the atmosphere where aerosols can still be present, this fact represents a problem when the solar background must be removed from the ceilometer signal. The normal procedure of solar background removal consists of subtracting from the ceilometer signal the median value of the signal itself over the last range bins (*far range*). This is only possible when the *far range* is not contaminated by aerosols or clouds. In order to overcome this problem a new technique of background removal depending on $VAR(S)$ has been developed and applied to each profile. $VAR(S)$ is calculated within spatial windows of 120 m to 1600 m width (in steps of 120 m) and computed for all range bins between 390 m and 14970 m. The background corresponds to the median value of $S$ over an optimal window. The optimal window's position is the one minimizing the average of its $VAR(S)$ values. The optimal window's width is the one corresponding to the 75[th] percentile of the $VAR(S)$ values at the optimal window's position. If it is true that the background correction needs more attention when measuring at tilted angle, a clear advantage related to in the  slant path is that once the ceilometer's beam reach the JFJ at 4.8 km the received signal is already in the full overlap region.

## 4    PathfinderTURB

PathfinderTURB adds a variance criterion to the original pathfinder scheme to retrieve the Continuous Aerosol Layer (CAL) and the CBL. The uncertainty related to the retrieval of the CBL and CAL is minimized by using the geodesic approach, which also allows a better adaptability of the algorithm to complex topography normally characterized by multiple aerosol layers. In the framework of de Bruine's work, the pathfinder technique was applied to the measurements of the tall-tower at the Cabauw site in the Netherlands and successfully validated by radiosonde (RS) data. Compared to other algorithms, pathfinder (and PathfinderTURB) can solve directly the attribution problem by building a time series of CBL (and CAL) heights using the geodesic approach between adjacent points (minimization of the cost function). PathfinderTURB has been applied to the ceilometer data at PAY and KSE.

## 4.1 Calculation of the Top of Continuous Aerosol Layer

The CAL is defined as the uninterrupted aerosol region along the backscatter profile starting from the ground and reaching the first discontinuity in the aerosol distribution. The top of the CAL (TCAL) is defined as the height of the retrieved discontinuity. The criteria to define the CAL are the following (see also Supplement S4):

1. *Signal condition*: the total (aerosol plus molecular) attenuated backscatter is larger than a threshold *Th* that depends on the purely molecular backscatter profile at the ceilometer's wavelength.

2. *SNR condition*: the signal-to-noise ratio (SNR) is larger than 0.6745.

Over flat homogeneous terrain, the TCAL usually corresponds to the top of the RL during night and to the height of the CBL during day. In complex and mountainous terrain, during daytime, the TCAL corresponds rather to the top of the so-called *injection layer*. The injection layer has been defined by Henne et al. (2004) as the layer formed by injections of CBL air at higher levels. The injections are engendered by thermally-driven converging slope winds along the topography reaching higher than the average in-valley CBL top. In contrast to the CBL, the injection layer is only sporadically mixed and indirectly connected to the surface. The *SNR condition*, imposes that the SNR is larger than the 1-sigma value of the signal noise. In other words, because the background signal (dark current plus stray light) is range-independent and is considered to be Gaussian-distributed, the backscatter signal is considered noisy when it lies within the 50%-confidence interval of the background signal. The noise is calculated in the far-range of the total signal. If the *SNR condition* is included, the retrieved TCAL can be shallower compared to when only the *signal condition* is taken into account. That happens especially during daytime when the SNR drops below the value 0.6745 already at low altitudes due to the enhanced solar background. In cases like this we cannot speak anymore of TCAL, but rather of maximum detected range. When clouds are present, also the height of the first cloud layer detected by the ceilometer combined with the heights obtained by the *SNR condition* and *signal condition* determine the TCAL.

## 4.2 Calculation of the Convective Boundary Layer Height

For a given day, the temporal evolution of the ceilometer signal is a matrix in time and space. Each column of the matrix represents a profile at time *t* and constant range resolution. The noise level is calculated from the photon counting signal using the method described by Morille et al. (2007). The signal is smoothed in space and time at resolutions of 30 m and 1 minute at PAY and, to compensate the reduced range due to the slant path, of 45 m and 2 minutes at KSE, respectively. We provide here a description of the main selection criteria and the main assumptions on which the CBL retrieval by PathfinderTURB is based. Further details about the algorithm, including the calculation of the atmospheric variability (signal variance) and of the turbulence-enhanced zones, and the mathematical steps leading to the expressions of the measured variables are given in the Supplement (S1, S2).

### 4.2.1 Lower altitude limit

Close to the ground, for most of the industrial bistatic ceilometers, the overlap between the transmitter and receiver is close to zero. In this region, called blind region, the returned signal is extremely weak, dominated by the noise and it oscillates around zero. It is thus not possible to retrieve the CBL height (CBLH) in this region

(low clouds or fog detections are however possible). Above this region, the overlap increases until becoming complete and the noise component becomes negligible compared to the signal at least within aerosol layers. A positive gradient is then expected at the transition between the blind region and the region above. We thus define the lower altitude limit, *minH,* as the first range where the transition from a zero to a positive gradient occurs and we impose *minH* not to be higher than 350 m (where the overlap of the ceilometer is normally sufficiently large to allow physical measurements).

During the morning and until the end of the afternoon, the CBLH exceeds the height *minH* due to its convective growth. An additional lower limit for the altitude is $minH_{TURB}$, which marks the onset of turbulence starting from the ground. Turbulence is calculated based on the temporal variation of the LIDAR signal for each $z$-level due to the atmospheric variability. The lower altitude limit *minH* is replaced by $minH_{TURB}$ whenever the latter is higher than the former. The selected minimum limit is called *liminf* in Figure 1.

### 4.2.2    Upper altitude limit

Different criteria are defined to calculate the upper altitude limit, *maxH*. These criteria are based on the a priori knowledge of the climatological CBLH value at a specific site (*climatological limit*) and on the retrieval of other aerosol and cloud layers that contribute to determine the actual CBLH. These layers are: the TCAL, the cloud base height (*cloud limit*), and the mixing discontinuities (strong negative and positive gradients). The minimum altitude amongst the three limits determines the upper altitude limit, *limphys*, shown in Figure 1.

*Climatological limit*

A climatological limit can be set based on visually-inspected ceilometer data from previous years and on model-simulated CBLH. The climatological limit depends on the site, and consists of a maximal CBLH value kept constant during the early morning, a maximal mean growth rate until the onset of the afternoon decay and a maximal CBLH value kept constant after the convective growth . For the PAY site, the period called "early morning"  starts at sunrise and ends 2.5 hours (3 hours at KSE) after sunrise. This period to the time-shifted onset convective plume and is assumed constant through the year. The afternoon period is considered to end at sunset. For our study we used the limits 1500 m a.s.l, 3000 m a.s.l and 1 km/h for PAY and 3069 m a.s.l., 4069 m a.s.l and 1 km/h for KSE for morning maximum CBLH, afternoon maximum CBLH and maximum mean growth rate, respectively.

*Cloud limit*

Two types of clouds are considered: CBL clouds and non-CBL clouds. All cloud information (number of cloud layers, cloud base, cloud depth) are provided by the ceilometer's standard outputs. A CBL cloud is defined as a cloud detected by the ceilometer in the first (lower) layer, whose vertical depth is less than 500 m and whose top (cloud base + depth) is lower than the site-specific climatological CBLH limit set beforehand. This criterion is purely mathematical as the cloud depth provided by the ceilometer just gives the depth of the not-totally attenuated part of the signal and not the real depth.

*Strong negative and positive gradients*

Strong positive or negative gradients indicate discontinuities in the vertical aerosol distribution and can then correspond to the CBLH. Strong positive gradients indicate normally a change from an FT region to an aerosol layer or a cloud base or from a CBL region to a cloud base. Strong negative gradients correspond to a signal drop between two adjacent gradient points of 25% (only 15% during the early morning period due to the still present RL above the forming CBL), whereas strong positive gradients correspond to a signal gain between two adjacent gradient points of 15% (only 5% during the early morning period due to the optically thin fog layer often lifted above the forming CBL).

### 4.2.3 Growth rate

Once the validity of the limits is accepted (e.g. the lower limit not exceeding the upper limit), the limits are recalculated back in time from 23:59 to 00:00, imposing a growth rate of $\pm 0.625 m/s$ between two time steps (i.e. $\Delta z < 37.5$ m at PAY and $< 75$ m at KSE). This growth rate is larger than the climatological growth rate of 1 km/h, because it allows larger jumps over shorter time steps in order to account for the convective dynamics, e.g. updraft and downdraft cycle.

### 4.2.4 Weights

At each time step $t$, a weight function $\Omega$ defines the "cost" of attributing the CBLH at the altitude z . The weights are calculated by PathfinderTURB as the product of the gradients weights and the variance weights. An offset is added to make the weights positive:

$$\Omega(t,z) = \log 10\big(\Omega_{Grad}(t,z) + \Omega_{Var}(t,z)\big) + \big|\min\big(\log 10\big(\Omega_{Grad}(t_{all}, z_{all}) + \Omega_{Var}(t_{all}, z_{all})\big)\big)\big| \qquad (2)$$

The offset is calculated taking the absolute minimum of $\Omega$ over the whole day and at all altitudes. The value of $\Omega_{Grad}$ is given by the inverse negative of the signal gradient, $\nabla S$ . The weights corresponding to positive or zero values of $\nabla S$ are set to 1000 times the largest weights of the inverse negative gradient values so that the cost of choosing a positive gradient is extremely high. The value of $\Omega_{Var}$ is given by the inverse of the signal variance, $VAR(S)$ .

For the KSE site the weights are calculated without the contribution of $VAR(S)$ . In fact the $VAR(S)$ is large when $S$ becomes noisy and shows a larger maximum at the noisy ranges instead at the CBLH. Because at KSE the signal is measured along a slant path, the noise is higher at lower altitudes compared to a vertical measurement (statistically over the entire dataset the SNR < 3 already at 850 m a.g.l.). Already within the CBL, the value of $VAR(S)$ in eq. (2) could lead to an incorrect retrieval of the CBLH, for this reason eq.(3) is used instead .

$$\Omega(t,z) = \log 10\big(\Omega_{Grad}(t,z)\big) + \big|\min\big(\log 10\big(\Omega_{Grad}(t_{all}, z_{all})\big)\big)\big| \qquad (3)$$

4.2.5      Shortest path
The shortest path in a graph (the geodesic in the metric space defined by the weights) is calculated using the
Dijkstra's algorithm (Dijkstra, 1959) and is based on the original method described by de Bruine et al. (2016).
The $\Omega$-weighted graph is constructed using the signal profiles starting from sunrise (midnight at KSE) over
consecutive intervals of 30 minutes (overlapping at first and last time steps) till sunset (23:59:59 at KSE). Within
the lower and upper altitude limits, the graph only allows connections of one time step in the positive time
direction and of maximum 37.5 m (75 m KSE) in the altitude direction. Every shortest path starts at time $t_i$ when
the previous shortest path has ended. In case of failure of shortest path calculation, the corresponding time
window is skipped, and the next shortest path starts at $(t_i, z)$ corresponding to the first local minimum weight. At
the first time step (sunrise for PAY, midnight for KSE), the CBLH is set at the first local minimum weight and
constraint by the lower graph limit . The CBLH time series calculated after sunset at PAY is discarded.
4.2.6      Ratio quality check
The retrieved CBLH is checked for quality at each time step by a binary quality index (0/1), where 0 corresponds
to no CBLH detection. In case of rain or fog, the quality index is set to 0. For all the other sky conditions in
order to perform a quality check we calculate the ratio of the mean ceilometer signal over 150 m above the
CBLH to the mean signal over the 150 m below the CBLH. When the ratio is larger than 0.85 (i.e. the signal
drop is less than 15%), the quality is set to 0, otherwise is set to 1.
**4.3     Example of PathfinderTURB's TCAL and CBLH calculation**
The retrieval's procedure of TCAL and CBLH can be summarized in three phases: pre-processing of $S$, CBLH
and TCAL retrieval, and quality-check. In the pre-processing phase, $\nabla \log_{10}(S)$, *liminf* and *limphys* are
calculated. In phase two, the time series of the range-restricted $\nabla \log_{10}(S)$ is transformed into a weighted
graph and the CBLH is determined as the geodesic calculated using the Dijkstra's algorithm (Dijkstra, 1959)
within predefined successive subintervals during the temporal interval between sunrise and sunset. Finally, in
phase three, the quality of the CBLH retrieval is assessed using the ratio quality check.
Based on equations (2)-(3), the geodesic can be calculated in the metric space defined by the weights.
PathfinderTURB calculates a line connecting the $(t_i, z)$ pairs that minimize the cost function defined by the
weights. The connecting line is the geodesic and has the property to strongly reduce the occurrence of unphysical
jumps between different layers when boundaries disappear or reappear due to real atmospheric dynamics.
PathfinderTURB uses $VAR(S)$ in addition to $\nabla S$ in order to solve the attribution problem in a more physical
way identifying regions characterized by large values of $VAR(S)$ and using it to retrieve the CBLH. The
CBLH is attributed to a layer's boundary in $(t_i, z)$ when this point minimizes the cost function, i.e. minimizes the
term $COST = \Omega_{Grad} \times VAR(S)^{-1}$. In this way, the influence of artificial and static aerosol gradients, present
in some models of ceilometers and due to an incorrect overlap correction, is largely reduced. The different steps
of the PathfinderTURB algorithm are illustrated in Figure 2, in four time series (panels a, b, c and d) for the case
of 15 July 2014 in PAY.
In Figure 2a, the logarithm of the range-corrected signal is displayed. The cloud base height (CBH) is directly
provided by the ceilometer manufacturer's software and displayed in grey throughout all panels. The TCAL
(shown in green) is the combination of the altitudes determined by applying the *signal condition* and the *SNR*
*condition* (sect. 4.1) plus the height of the first cloud layer. The *signal condition* and the CBH play a critical role
in this example. The development of the CBL is limited in altitude by the TCAL, but it can also be limited below
the TCAL by the other limits contributing to *limphys* (sect. 4.2.2). The *limphys* is the minimum height amongst
the climatological, the TCAL, the CBH and that of strong negative and positive gradients (indicating mixing
discontinuities). During the period 02:00-03:30 UTC, *limphys* (magenta) was determined by strong positive
gradients at about 1500 m a.s.l.; during 20:00-24:00 UTC and at about 1750 m a.s.l.by strong negative gradients.
In Figure 2b, the $VAR(S)$ is displayed. $VAR(S)$ is calculated using spectral analysis, more precisely is the
result of integrating the spectrum of band-pass filtered, one-hour-long *S*-time series at each altitude (as in Pal et
al., 2013). The band-pass filter aims at removing mesoscale and noisy fluctuations so that only fluctuations due
to short-lived aerosol load variability are taken into account (Supplement S2). The lower altitude limit (*liminf*) is
calculated based on the $VAR(S)$ value, and displayed in magenta. The next region of enhanced $VAR(S)$
above *liminf* corresponds to the EZ at the top of the CBL, the fact of preventing the retrieval to include the
region [0- *liminf*] helps locating correctly the CBLH.
In Figure 2c, the weights $\Omega(t,z)$ are displayed. Based on equations (2), (3) the CBLH-path follows the deep-
blue regions corresponding to a minimum in $\Omega(t,z)$. The path can only follow the positive time direction and
altitude changes are limited to 0.625 m/s. The CBLH is characterized by a drop in the aerosol concentration
(large negative $\nabla S$) and high entrainment activity (large $VAR(S)$), which corresponds to minimum $\Omega(t,z)$.
In Figure 2d, a final overview is given, with the retrieved CBLH (black line) displayed on top of the log10(*S*)
time series, together with the TCAL and the CBH.

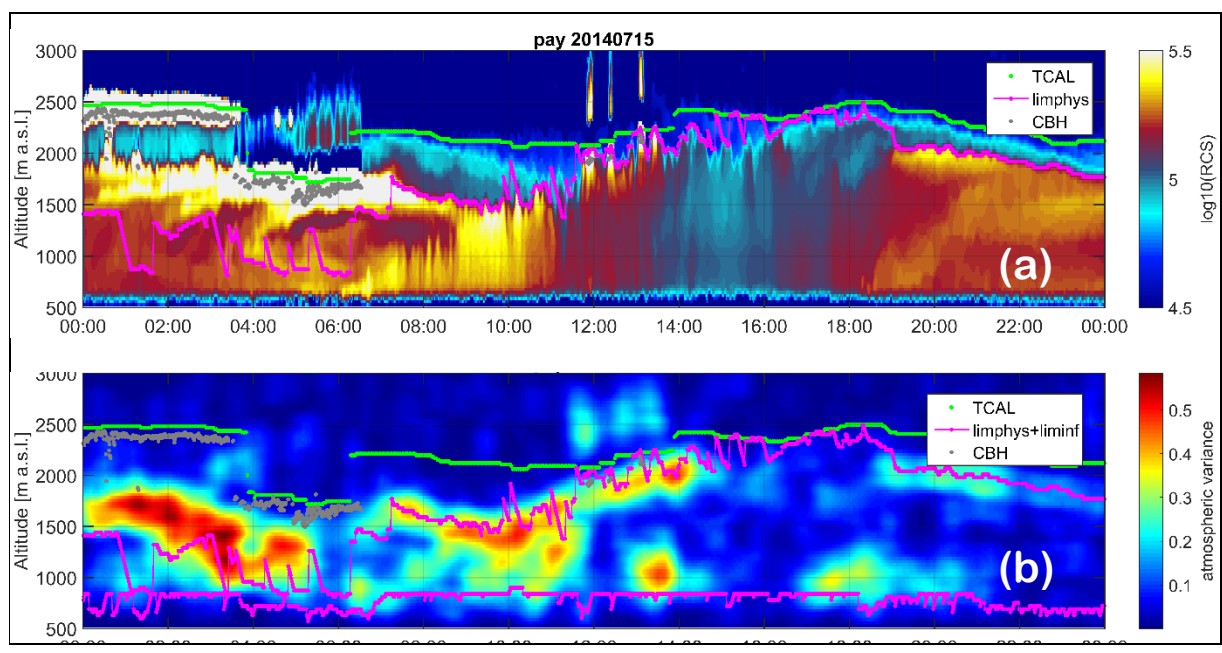

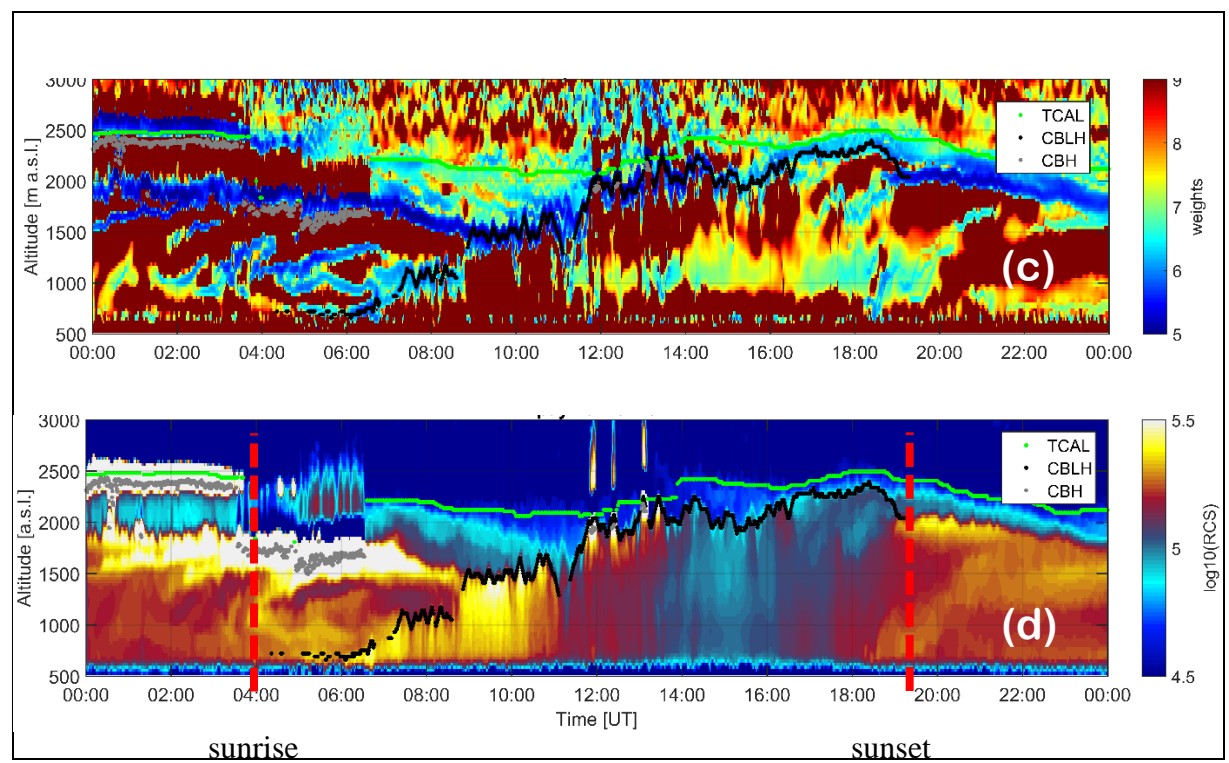

Figure 2. Time series of the different processing steps of the PathfinderTURB for 15 July 2014 at PAY. Panel a)
time series of $\log10(S)$ with superimposed TCAL, CBH and *limphys* for altitude. Panel b) time series of
$VAR(S)$ with superimposed CBH, *limphys*, *liminf* for altitude. Panel c) time series of $\Omega(t,z)$ with
superimposed TCAL, CBH and retrieved CBLH (geodesic from sunrise to sunset). Panel d) time series of
$\log10(S)$ with superimposed TCAL, the CBH and the retrieved CBLH.
## 5   PathfinderTURB validation at Payerne
Although, gradient-based algorithms are easy to implement for automatic operations, the layer attribution
remains the main source of uncertainty of the retrievals. For methods based on aerosol gradients the visual
identification of the correct gradient by human expert still solves the attribution problem with the least
uncertainty. Therefore, PathfinderTURB is validated here against independent detections by human experts as
well as against the bulk-Richardson method applied to co-located radiosonde profiles. The aim of the validation
is to create an as-accurate-as-possible reference without selecting only golden-cases, but filtering out those cases
when fog and precipitation prevent to define the CBL.
### 5.1   Comparison with Human expert CBLH retrieval
A graphical user-interface has been developed for the human experts to detect the CBLH manually by clicking
on the time-height cross section of *S*. Auxiliary information are available from the interface about $\nabla S$;
$VAR(S)$ (over 10 minutes); sunshine duration, vertical heat flux at the surface; trend of hourly-averaged
surface temperatures $\Delta T$; hourly stability index (as defined in Pal et al., 2013); sunset/sunrise time; estimations
of the CBLH based on the Parcel method (PM, Holzworth, 1964) and the bulk Richardson method (*bR*,

Richardson, 1920) from continuous remote sensing instrumental data (Microwave Radiometer, Wind Profiler, Raman LIDAR), and twice daily radiosounding data (at noon and at midnight). The experts perform a manual detection of the entire daily cycle with the support of all the ancillary data and information.

Four experts from the remote-sensing division of MeteoSwiss have processed one year of data (2014) of the PAY CHM15k. The guidelines and the criteria of the manual CBLH detection are provided in the Supplement (S5).

### 5.1.1 Analysed dataset

We compared the detections by three experts (*test group*) against one expert that acted as *reference*. For the year 2014, the analysed days were the $5^{th}$, $10^{th}$, $15^{th}$, $20^{th}$, $25^{th}$ and $30^{th}$ of each month and the whole months of January, March, July and October. The *test group* analysed the $5^{th}$, $10^{th}$, $15^{th}$, $20^{th}$, $25^{th}$ and $30^{th}$ of each month. Once the missing data (due to instrument disruptions) and fog or precipitation days had been removed from the dataset, the total number of days analysed was 174. Covering an entire year, the database inspected by the *test group* and the *reference* is comprehensive in terms of diverse synoptic conditions, sunshine duration, cloudiness and season. The *S*-profiles were analysed by the *test group* separately and with no possibility to influence each other's choice. The detections made by the *reference* and those made by the *test group* were compared at each time step so that a matching procedure was established between a CBLH point in the *reference* and the *test group*'s detections for the same time step. Only the CBLH points that matched in time were retained for the comparison. If needed, the *test group* detections were linearly interpolated in order to match exactly the time vector of the *reference*. When comparing the *reference* with all the *test group* detections the two datasets showed an excellent agreement, with a coefficient of determination of 0.96 (total of 5097 points over 140 days) and RMSE of 92 m. Nevertheless, some large difference (> 500 m) in the CBLH detections occurred in less than 3 % of all cases. In general, discrepancies occurred when there was more than one layer that could be reasonably followed as CBLH, for example when an advected aerosol layer entered the profile and got mixed inside the CBL or during the often ambiguous separation between the RL and the decaying CBL in the afternoon after the convective peak.

### 5.1.2 PathfinderTURB validation against the expert consensus

After applying the ratio quality check (sect. 4.2.6) to the PathfinderTURB retrievals, the total number of the accepted retrievals covers 34720 minutes of the 43914 minutes obtained by the manual detections, i.e. 79% of the human expert consensus. The ratio quality check of PathfinderTURB removes about the 20% of the retrievals because of weak gradients at the level of the retrieved CBLH. The validated PathfinderTURB retrievals are distributed over the same number of days (i.e., 135) during the year 2014; the top panel of Figure 3 shows the density scatter plot of the CBLH values obtained at PAY by the (consensus) manual detections versus PathfinderTURB. The boxplot along with the histogram shown in the bottom panel display the differences between the two data sets. A coefficient of determination of 0.96, an RMSE of 76 m and an interquartile range of the differences of 96 m are obtained. The median and mean differences are 27 m and 41 m, respectively. The overestimation is largest during the second half of the afternoon (not explicitly shown here), when PathfinderTURB tends to follow the top of the residual layer instead of the decaying CBL. Furthermore, the

error is smaller than 500 m for 98.6% of the PathfinderTURB retrievals, and 92% of the retrievals have a relative
error (with respect to the manual CBLH) smaller than 10%.

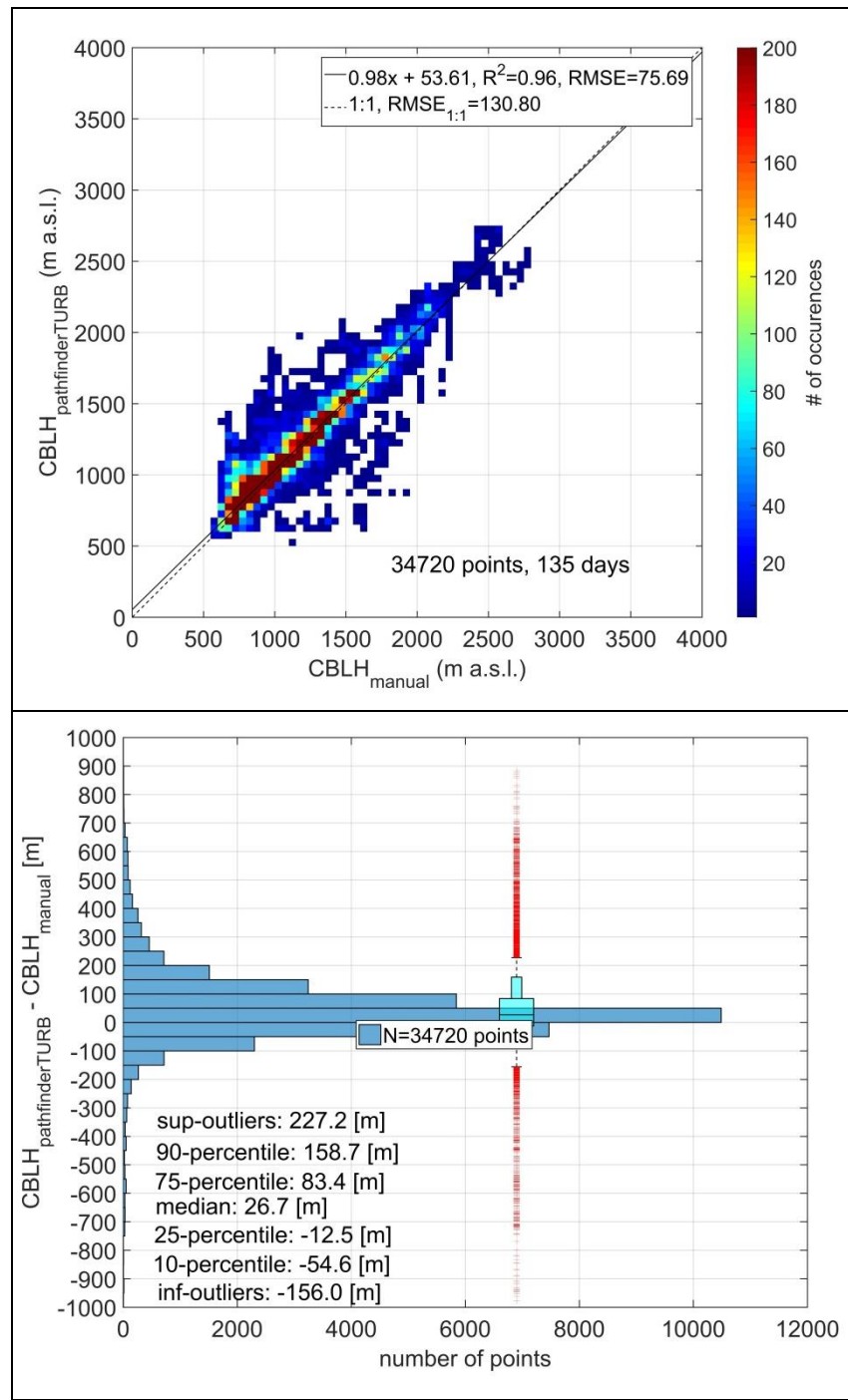

Figure 3. Density scatter plot of CBLH$_{PathfinderTURB}$ vs. CBLH$_{manual}$ (top panel). Boxplot and histogram of the
difference between the PathfinderTURB and manual datasets (bottom panel).
The comparison shows that PathfinderTURB is robust and that can address the attribution problem adequately.
Although PathfinderTURB combines both gradient and variance methods to improve the correctness of the
retrieval in different atmospheric conditions, the retrieval's uncertainty grows larger during the afternoon due to

the decay of convection before sunset, the weak turbulence and the lack of well-marked aerosol gradients. During this period, temperature or vertical wind variability profiles may provide more valuable information than ceilometer profiles.

## 5.2  Comparison with radiosonde-estimated CBLH

The PathfinderTURB retrievals of the CBLH were compared to the retrievals from two methods based on the thermal structure of the atmosphere: the PM and the $bR$. The PM defines the CBLH as the height to which an air parcel with ambient surface temperature can rise adiabatically from the ground, neglecting other factors (entrainment/detrainment, advection, subsidence, air humidity). It relies on profiles of potential temperature ($\Theta$) and therefore requires vertical profiles and surface values of temperature ($T$) and pressure ($p$). In Payerne, $\Theta$ profiles are generated every 10 minutes by a microwave radiometer (MWR), and at noon and midnight also by RS. The bulk Richardson number ($Ri_b$) is a dimensionless parameter that can be seen as the ratio between the buoyancy and the wind-shear generated turbulence. The CBLH is determined as the first height where $Ri_b$ exceeds the critical threshold of 0.33 (unstable conditions) or of 0.22 (stable conditions). The required input values are the profiles of $\Theta$ and the wind. The stability conditions, essential for choosing the correct threshold value, are derived from the sign of the slope of the linear fit of $\Theta$ in the first 30 m. At Payerne, wind profiles are provided every 30 minutes by the wind profiler (WP), and at noon and midnight also by RS. We refer to Collaud Coen et al. (2014) for a more detailed description of the operational CBLH retrievals at Payerne using the $bR$ method.

PathfinderTURB is compared to the RS-based $bR$ retrievals of the noon CBLH during the year 2014. In order to increase the robustness of the $bR$ retrievals, the comparison is performed only when both $bR$ and PM retrievals are available. Based on the calculations of Collaud Coen et al. (2014) the uncertainty of the retrieved CBLH using both methods is on the order of ±50 to ±250 m for the midday peak of the CBLH. Within their uncertainty intervals, the two methods can then be considered providing the same retrievals when the difference between them is equal to or less than 250 m. For this reason only the retrievals matching closer than 250 m and with an uncertainty of less than 250 m have been retained for the comparison. That has resulted in a total of 175 days being considered. Of these 175 days only on 115 days PathfinderTURB could retrieve a valid CBLH. Thereafter and for simplicity, only the $bR$ retrievals will be used in the comparison with PathfinderTURB ($bR$ and PM pairs were always available for the considered 115 days) .

The median and mean difference between RS and PathfinderTURB CBLH values were 53 m and 41 m, respectively, indicating a slight overestimation of the $bR$ method with respect to PathfinderTURB. From the comparison we obtain a coefficient of determination of 0.85, a regression slope of 1.02 (Figure 4, top panel), an RMSE of 162 m and an interquartile range of the difference of 174 m, (larger than the spread observed in Figure 3). The distribution of the differences in Figure 4, bottom panel, has a Gaussian shape with slight positive offset values. About the 98% of the data have an error smaller than 500 m, and the 82% have an error smaller than the 10% (plus 100 m) of the CBLH retrieved by $bR$. In general, the correlation between PathfinderTURB (ceilometer-based) and the $bR$ retrievals (RS-based) is not as good as the one between PathfinderTURB and the manual retrievals (both ceilometer-based). For the comparison shown in Figure 4, it should be remembered that

the two methods rely on different physical processes, i.e. thermal structure of the atmosphere (RS) versus actual
state of mixing of the aerosols (ceilometer). A consequence of the different physical processes is the slight
overestimation of the *bR* method during the period from the end of morning till the beginning of afternoon, i.e.
when buoyancy-produced turbulence reaches a maximum. This is because the *bR* indicates the depth of the layer
where conditions are favourable for vertical mixing, whereas the aerosol gradient depicts the actual state of
mixing. By using the MWR data to evaluate the entire daily cycle (not only 12 UTC by RS) the comparison
PathfinderTURB vs *bR* shows that the *bR*-based CBLH rises generally faster than the aerosol gradient in the
morning. The decay of the *bR*-based CBLH occurs also generally faster than that of the aerosol gradient in the
late afternoon, resulting in *bR* retrievals lower than the PathfinderTURB CBLH retrievals. This is explained by
the fact that the aerosols remain suspended in the near-neutrally stratified air (transition from CBL to RL) and
that no detectable aerosol gradient forms at the top of the decaying CBL. The gradient remains thus at about the
same altitude as its midday maximum leading to a significant overestimation by PathfinderTURB. For this
reason, LIDAR and ceilometers using aerosols as tracers are not best suited to detect the CBL decay, but rather
the RL. Nevertheless, although at 12:00 UTC the *bR* still provides a slightly higher CBLH, the comparison
shown in Figure 4 proves a good agreement between *bR* and PathfinderTURB.

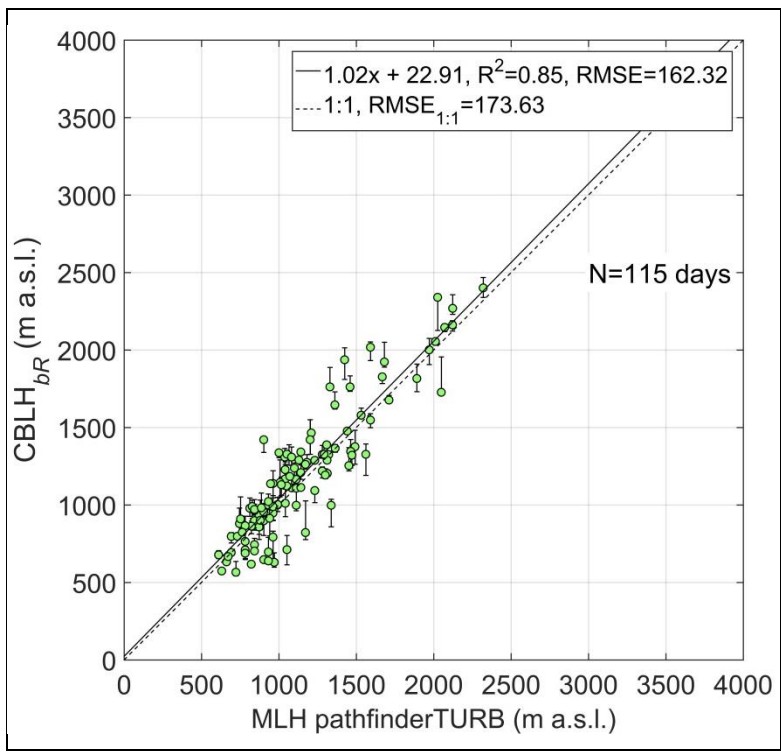

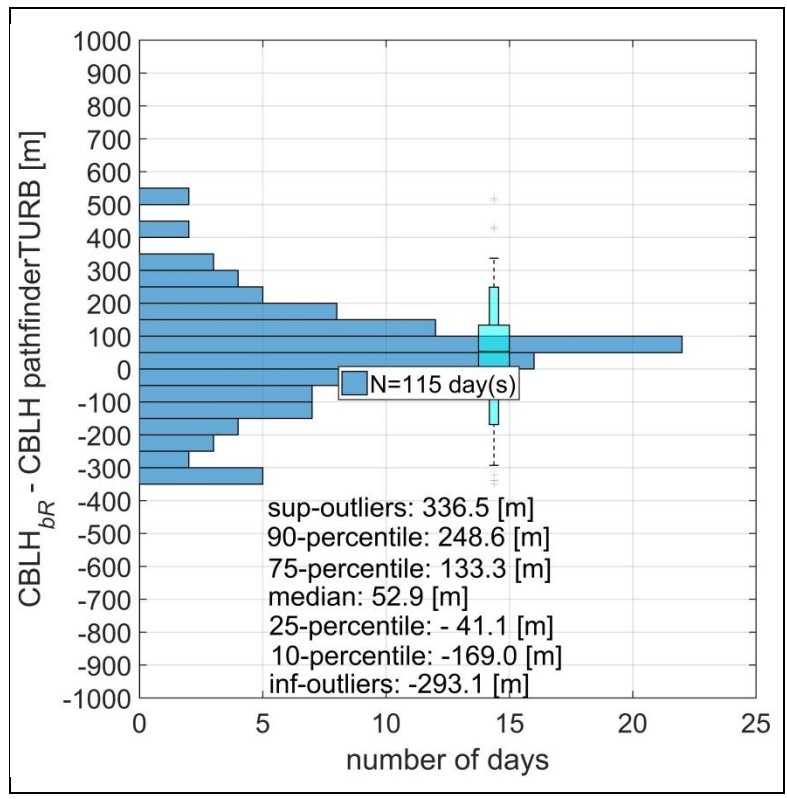

Figure 4. all data shown refer to 12:00 UTC. Top panel: scatter plot of CBLH$_{bR}$ vs. CBLH$_{PathfinderTURB}$. Bottom panel: boxplot and histogram of the difference between the *bR* and PathfinderTURB datasets

## 6    Measurements of CBL, CAL and aerosol properties at JFJ

Updrafts and downdrafts (initiated and sustained by solar radiation received at the surface) are the main vertical transport mechanism of the CBL air above the Swiss Plateau (Collaud Coen et al., 2011). Air lifted from a sunlit mountain slope is often warmer than the air at the same height over an adjacent valley even if the latter was lifted from the valley floor. Hence, next to the development of up-slope (anabatic) winds, thermals generated at a mountain slope may rise higher than those generated at the valley floor. When both the topography and the meteorological conditions are favourable, up-slope winds can develop and become strong enough to break through the CBL's capping inversion and inject CBL air into the FT immediately above the local CBL (LCBL) resulting in the formation of an aerosol layer (AL) above the CBL (Henne at al., 2004). This complex mountain circulation is characterized by dynamics occurring at different spatial scales (Figure 5). The AL or *injection layer* is a near-neutral, partly-mixed layer that is more diluted than the LCBL being the result of LCBL air mixed with FT air. The LCBL normally follows the topography (scale of a few kilometres), especially in the morning, and is often topped by a temperature inversion that marks the transition with the above AL. At its upper boundary, the AL does not follow individual valley or ridges, but follows the large-scale topography (few tens of kilometres) and can also be overlaid by a temperature inversion marking the transition with the FT (Henne at al., 2004, de Wekker, 2002). In his work, de Wekker (2002) concludes that in mountainous regions, the mixing layer height corresponds to the top of the AL rather than the top of the LCBL and he renames it "mountain mixing layer", because the AL depicts the height up to which particles can be transported by the various venting processes. The combination of the LCBL and the AL forms the CAL (Figure 5).

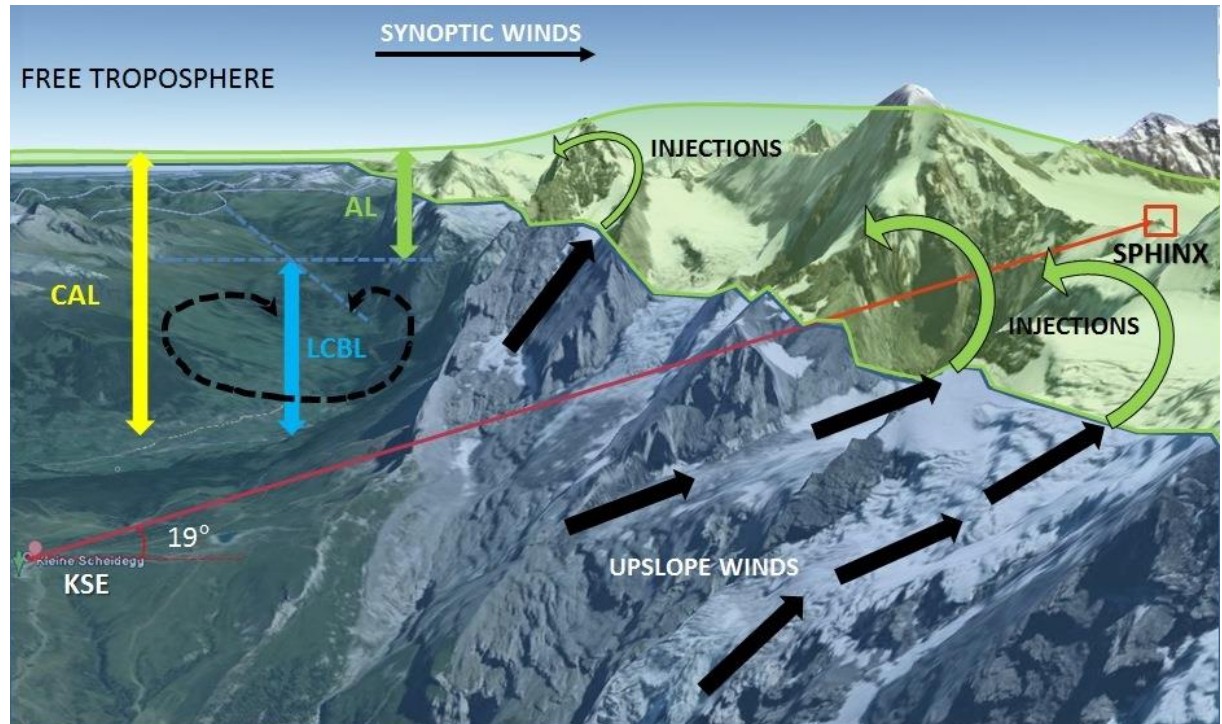

Figure 5. Schematic view of the daytime atmospheric structure and vertical pollution transport in and above the KSE site. The red line shows the CHM15k line of sight towards the Sphinx. The annotations denote the different thermal transport and mixing mechanisms of boundary layer air.

At the JFJ, aerosols and gases have been measured continuously since many years. Different sources and transport regimes towards the JFJ have been studied by many authors (e.g., Lugauer et al., 1998; Zellweger et al. 2003, Balzani Lööv et al. 2008, Henne et al. 2010, Collaud Coen et al., 2011; Collaud Coen et al., 2014; Herrmann et al. 2015) showing that the JFJ resides most of the time in the undisturbed ("clean") lower FT. Nevertheless and especially in summer, the JFJ is influenced by thermally-induced uplifted CBL air, it is also influenced by additional lifting processes such as frontal passages and Föhn flows (Zellweger et al., 2003, Ketterer et al., 2014). As observed by Zellweger et al. (2003) the thermally-induced transport of CBL air towards the JFJ occurs frequently during summer (~35 % of the time). The previous studies suggest that the direct contact of undiluted LCBL air with the in-situ instruments at the JFJ occurs only rarely and is limited to summer periods (e.g., Ketterer et al., 2014). Lugauer et al., (1998) provide a 9-year climatological analysis of the vertical transport of aerosols to the JFJ and the corresponding synoptic conditions. The thermally-induced transport is nearly absent in winter or under cyclonic conditions and it is strongest in summer under anticyclonic periods. During favourable conditions, the aerosol concentration increases at the JFJ during the afternoon with a peak at around 18:00 UT and the peak is stronger in northern synoptic wind than in southern because of the difference in upwind topography. Collaud Coen et al. (2011) found as well that the JFJ is mainly influenced by free tropospheric air masses in winter and largely influenced by the LCBL (also during the night) in summer during subsidence periods.

In order to understand the impact of the thermally-driven dynamics on the in-situ measurements at the JFJ and to quantify, by direct observations, the number of times that the LCBL and the CAL reach the JFJ throughout the year, the data from the CHM15k have been analysed using PathfinderTURB during the period August 2014 till

November 2015. PathfinderTURB has been adapted to use the CHM15k data along the slant-probing direction connecting KSE with JFJ. The adapted PathfinderTURB version does not use the $VAR(S)$ profiles to calculate the weights (eq. (3)), but solely to retrieve the first transition to the enhanced turbulence zone (see Sect. S2 in the Supplement). In fact, at close ranges, where the first transition to the turbulent region is usually found, the $S$ profile has a much higher SNR and $VAR(S)$ can be measured reliably. At KSE, the LCBL height (LCBLH) retrieved by PathfinderTURB, corresponds to the first discontinuity in the vertical mixing of aerosols and can be estimated also during nighttime.

## 6.1 Retrieval of aerosol layers at KSE and JFJ

The CHM15k detects the aerosols that form in the surrounding lower-altitude valleys (e.g., 1034 m a.s.l. Grindelwald, 566 m a.s.l. Interlaken) and that are transported above the KSE. Local generation of aerosols occurs only sparingly due to the reduced vegetation and the long periods of snow and ice cover. Nevertheless, when local aerosols production occurs, these can be transported through the ceilometer's field of view and eventually be transported up to the JFJ. The local aerosols generation and the advection from the surrounding valleys lead to different scenarios. During daytime, both TCAL and LCBLH can be detected, the LCBLH only during periods when the LCBL air is lifted into the ceilometer's field of view by convection. During nighttime, when there is no convection, only the TCAL can be detected (if it is present). The nocturnal TCAL can stem from the residual layer formed above the surrounding valleys. PathfinderTURB is based on the same retrieval principle during day and night, and so it looks for the first discontinuity in the uninterrupted aerosol region. For this reason and for simplicity we will refer to the retrieved nocturnal boundary layer as to LCBL even when the mixing is not due to convection, but rather to mechanical mixing from the surface and katabatic winds.

### 6.1.1 LCBLH retrieval

The seasonal-averaged daily cycles of the retrieved LCBLH and TCAL during spring, summer, autumn and winter are shown in Figure 6. During spring (Figure 6a), summer (Figure 6b) and (partially) autumn (Figure 6c), the LCBLH grows through morning till reaching a peak in the afternoon. In summer, the LCBLH has been retrieved by PathfinderTURB every day with only few exceptions. In Spring, (March-May), and in summer (June-August) the LCBL has reached the JFJ during 20 and 9 individual days, respectively. These occurrences lay above the 75 percentile of the LCBLH dataset and, hence, are not represented by the blue-shaded area in Figure 6a-b. From the systematic visual inspection and comparison of LCBLH time series at PAY and KSE, we can say that the LCBLH peak occurs later at KSE than at PAY. During the night, the LCBLH drops, due to the concurrent effects of aerosol gravitational settling, subsidence and katabatic winds, which result from radiative cooling of the surface triggering katabatic drainage flows. A likely explanation of the delay in the onset of the LCBL and of the afternoon peak at KSE is that due to the nighttime katabatic winds driving FT air down into the valley underneath. Depending on the season, these winds can continue to blow for few hours after sunrise (especially from the shaded mountain side) and work against the formation of the LCBL. The LCBLH temporal evolution follows the classical shape of a growing convective boundary layer like over flat terrain, but the growth and the duration of the LCBL occur over a shorter period. This is consistent with the delayed onset of the LCBL due to the persisting katabatic winds in the first hours of the morning and the earlier weakening of

convection due to the shading effect of the surrounding mountains and the afternoon onset of the katabatic winds. This phenomenon is particularly enhanced during winter when the solar irradiance is at its minimum and the katabatic winds tend to suppress LCBL during most of the time.

In autumn (September-November), the LCBLH shows a less pronounced daily cycle than in spring and summer, this is probably due to the fact that the vertical transport of aerosol-rich air is reduced by the stabilization within the lower troposphere during this period (Lugauer et al., 1998).

In winter, (December-February) PathfinderTURB could retrieve only few LCBLH because of the very stable meteorological conditions, the reduced convection and the prolonged snow and ice cover limiting the aerosol production at KSE and the surrounding valleys. For this reason the seasonal-averaged daily cycle in Figure 6d does not show any particular pattern of the LCBLH, mainly due to the very low retrieval counts.

All occurrences of when the LCBLH and TCAL have reached the JFJ during the different months are listed in Table 1.

### 6.1.2 TCAL retrieval

During spring and autumn, the daytime TCAL evolution is correlated with the LCBLH especially in spring during the first hours after sunrise (convective growth) and until the afternoon peak. The nighttime evolution of the TCAL in spring and autumn also shows a correlation, although weaker, with the LCBLH. In summer, the TCAL does not show any significant correlation with the temporal evolution of the LCBLH during day or night. During winter, the TCAL shows no correlation with the LCBLH. Despite an overall absence of a daily pattern of the winter LCBLH, the TCAL shows a clear outline during the period 00-10 UTC. This bimodal pattern with higher TCAL during the first part of the day could be explained by the process of dissipation of the CAL caused by the wind shear along the line of sight connecting KSE and JFJ when the solar irradiance modifies the wind dynamics during the central hours of the day.

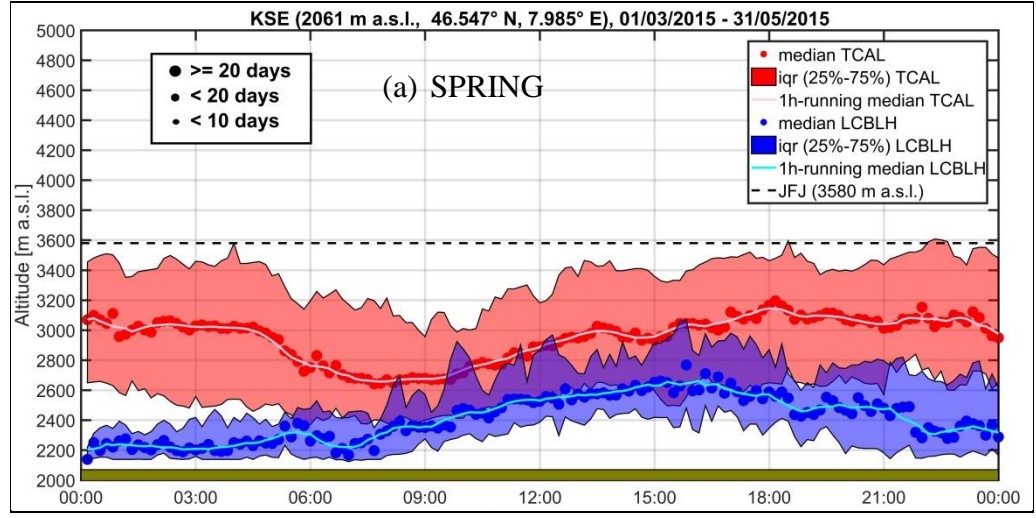

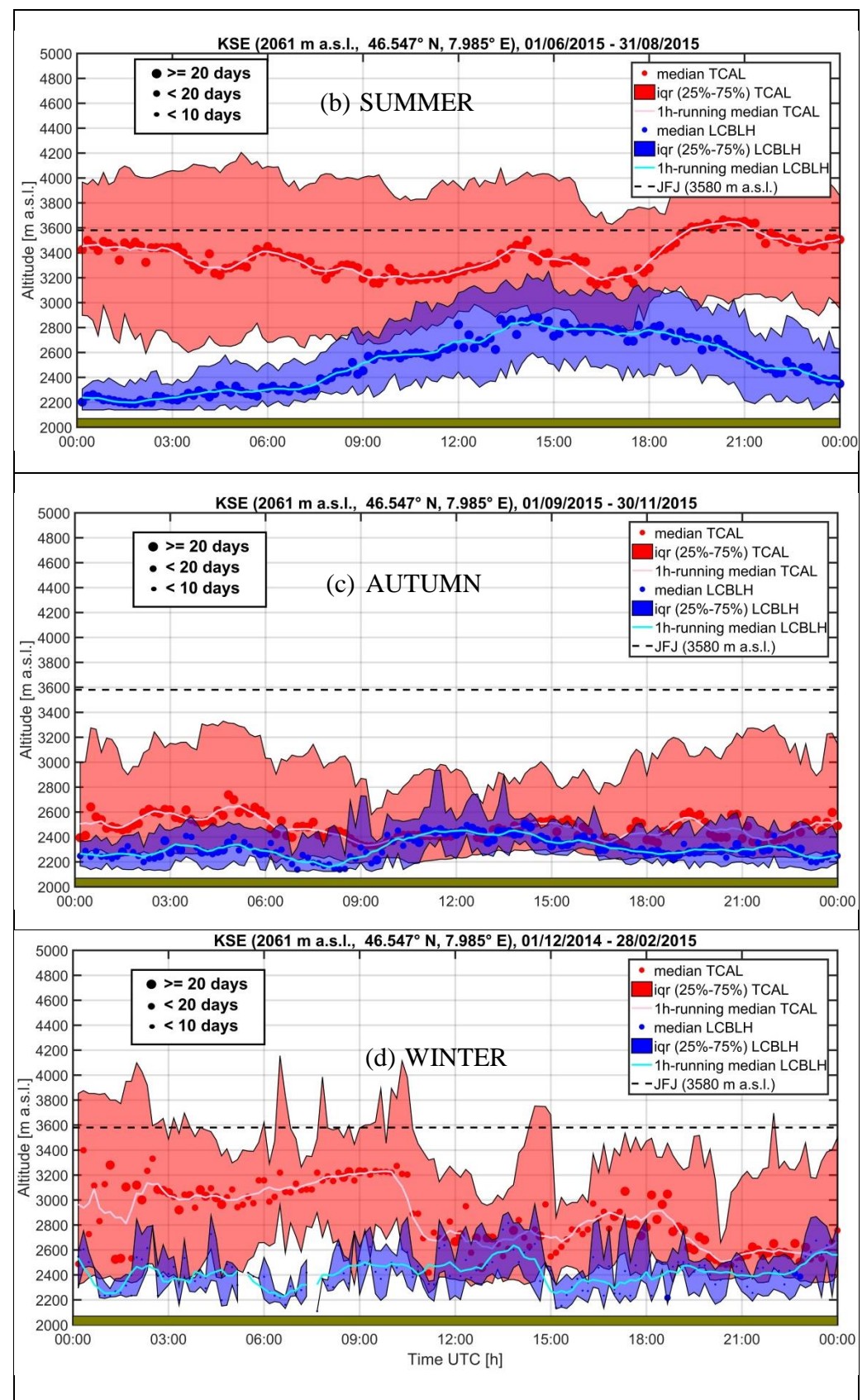

Figure 6. Season-averaged daily cycle of the TCAL (red dots) and of the LCBLH (blue dots) at KSE. The size of the dots corresponds to the number of measurements available in each temporal bin. Panel a) spring (March to May); panel b) summer (June to August); panel c) autumn (September to November); panel d) winter (January to February). Shaded areas show the 25%-75% inter-quartile range (iqr) for LCBL (purple) and TCAL (red). The altitude of JFJ is indicated by the black-dashed horizontal line.

### 6.1.3    Occurrence frequency of LCBL and CAL reaching JFJ

Table 1 shows, for each month during the studied period, the number of hours (cumulative 2-minute data points over the month), the number of days (number of days with at least one data point), and the percentage of time (time when the JFJ was inside LCBL or CAL as a percentage of the total time when the retrievals existed). On the left-hand side of the table, we show the statistics corresponding to when the JFJ is reached by or embedded into the LCBL, on the right-hand side we show the statistics corresponding to when the JFJ is either into the LCBL or the CAL (LCBL + AL). The statistics show that during winter (light grey rows in Table 1) the aerosol measurements at the JFJ are never directly influenced by the LCBL air, which remains constantly below the JFJ. Moreover, the total duration of time when PathfinderTURB has detected the LCBL rising above KSE (but not touching the JFJ) during winter accounts for no more than 65.52 hours. On the other hand, the CAL reaches the JFJ about one fourth of the time (21.23%), which corresponds to a duration of 109.44 hours (distributed over 26 days). The remaining three quarters of time (78.77%), corresponding to a duration of 406.32 hours, the JFJ is situated in the FT, i.e. the in-situ measurements are characterized by background (molecular) conditions. Although it is impossible to establish the exact origin of the air in the AL (i.e., the injection layer), we can speculate that winter AL is composed of aerosols originating from long-range transport and synoptic-scale lifting, rather than LCBL injections.

During summer (dark grey rows in Table 1) the situation changes significantly with the LCBL reaching the JFJ during the 3.63% of time, corresponding to 34.56 hours (distributed over 20 days).

Table 1: statistics of frequency of LCBL and CAL reaching or embedding the JFJ.

| JFJ inside LCBL | | | | JFJ inside CAL | | | |
|---|---|---|---|---|---|---|---|
| **Date** | **Hours** | **Days** | **%** | **Date** | **Hours** | **Days** | **%** |
| 09/2014 | 4.87 | 2 | 1.94 | 09/2014 | 149.03 | 18 | 23.882 |
| 10/2014 | 8.00 | 4 | 5.81 | 10/2014 | 88.70 | 18 | 16.29 |
| 11/2014 | 1.67 | 3 | 4.20 | 11/2014 | 72.23 | 13 | 24.49 |
| 12/2014 | 0.00 | 0 | 0.00 | 12/2014 | 43.30 | 9 | 26.67 |
| 01/2015 | 0.00 | 0 | 0.00 | 01/2015 | 33.53 | 10 | 21.76 |
| 02/2015 | 0.00 | 0 | 0.00 | 02/2015 | 32.70 | 7 | 16.41 |
| 03/2015 | 0.2 | 1 | 0.12 | 03/2015 | 45.77 | 13 | 10.24 |

| | | | | | | | |
|---|---|---|---|---|---|---|---|
| 04/2015 | 5.67 | 3 | 3.59 | 04/2015 | 80.73 | 15 | 14.82 |
| 05/2015 | 5.43 | 5 | 2.21 | 05/2015 | 114.07 | 17 | 22.83 |
| 06/2015 | 0.50 | 2 | 0.16 | 06/2015 | 174.60 | 24 | 29.30 |
| 07/2015 | 18.60 | 12 | 5.61 | 07/2015 | 380.47 | 28 | 56.49 |
| 08/2015 | 15.50 | 6 | 5.17 | 08/2015 | 217.63 | 19 | 36.34 |
| 09/2015 | 0.97 | 2 | 0.51 | 09/2015 | 56.30 | 12 | 11.50 |
| 10/2015 | 0.00 | 0 | 0.00 | 10/2015 | 19.87 | 6 | 5.53 |
| 11/2015 | 0.2 | 1 | 0.36 | 11/2015 | 4.10 | 2 | 1.42 |

Although the relatively low percentage may induce to think of a marginal effect, the striking parameter is that
during summer the undiluted, aerosol-laden air of the LCBL is able to reach the JFJ (and potentially strongly
affect the in-situ measurements of particles concentrations and of their optical properties) on 20 different days.
With regard to the frequency and duration when the CAL has reached or embedded the JFJ, the statistics are
even more remarkable with the 40.92% of the time or 772.8 hours distributed over 71 days. Also for the summer
statistics, no quantitative conclusions can be drawn about the origin and type of the aerosols inside the AL. The
aerosols could be locally emitted and injected into the AL or transported over regional or continental scales and
could have been formed secondarily in the AL (Bianchi et al., 2016). In any case, the convective conditions
occurring frequently during the summer suggest a significant mixing of the LCBL air into the FT forming the
AL. Also the measurements by the in-situ instrumentations at the JFJ show that the absorption coefficient
(indirectly proportional to the black carbon concentration) is largest during the summer period.
As mentioned in the previous sections, these results are in agreement and confirm the indirect measurements and
model simulations done in the previous works, especially those by Zellweger et al. (2003), Collaud Coen et al.
(2011), Ketterer et al. (2014) and Herrmann et al. (2015). Here, and for the first time, the occurrence of the
convective (LCBL) and injection (AL) layers reaching directly the in-situ instrumentation at the JFJ has been
statistically analysed based on one year of data. The big advantage of applying PathfinderTURB to the
ceilometer profiles is to have an automatic retrieval of the LCBLH and the TCAL directly at the JFJ. That allows
to have real values of  LCBLH or TCAL at the JFJ and not to use detections  made in an atmosphere  distant
many kilometres from the JFJ. Moreover, not co-located measurements require stringent homogeneity conditions
of the atmosphere between the point where the LCBLH and TCAL have been detected and the JFJ.
**6.2    Comparison with in-situ instrumentation**
Figure 7 shows the relation that exists between the daily maximum of the LCBLH retrieved by PathfinderTURB
and the corresponding (in time) absorption coefficient, $\alpha$, at 637 nm measured by the MAAP at the JFJ. The
vertical red dashed line shows the altitude of the JFJ, each box collects all the LCBLH retrievals within 400-m of
vertical span and the corresponding values of $\alpha$. In each box the number of LCBLH- $\alpha$ pairs is indicated by N
and the median of each box is connected by the black dashed line to show the median trend. The data in the box-
plot are from all seasons, in order to maximize the number of occurrences and increase the statistical significance
of the trend.

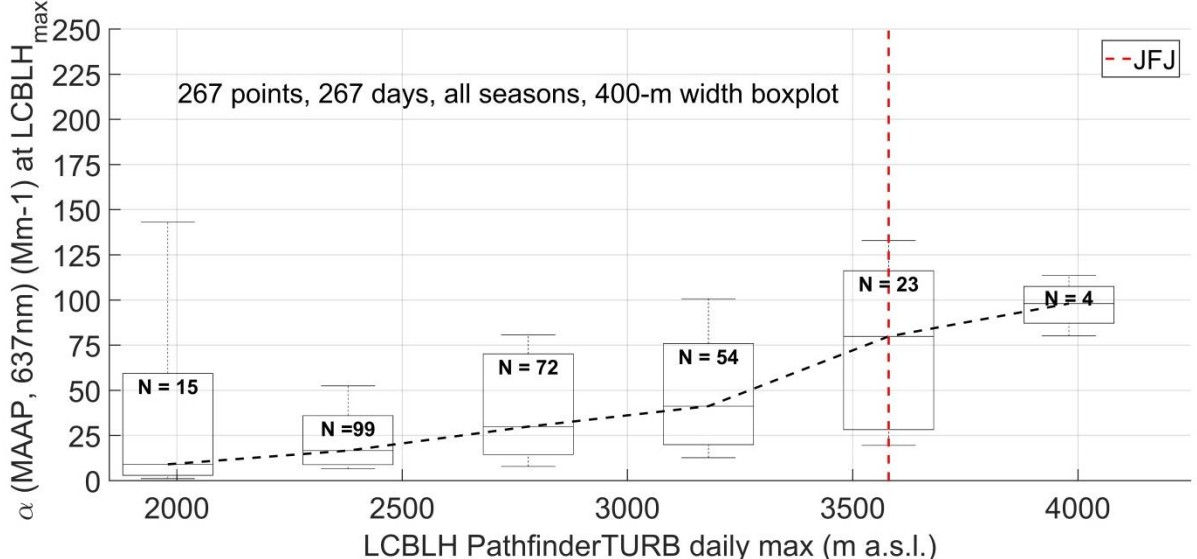

Figure 7. Box-plot showing the relation between the absorption coefficient at 637 nm measured by the MAAP at
the JFJ vs the LCBLH retrieved by PathfinderTURB for the period September 2014 to November 2015.
A linear median trend characterised by a small slope could be fitted to the LCBLH-α pairs for LCBLH lower
than the JFJ (2000-3380 m). For these range of altitudes the LCBL grows deeper getting closer to the height of
the JFJ. The injections of LCBL air into the AL (embedding the JFJ) are then more likely to occur when the
LCBLH reaches its maximum injecting LCBL air past the in-situ sensors with a resulting higher value of α. As
soon as the LCBLH reaches the JFJ, the injections into the AL reaching the in-situ instrumentation become more
important and this is shown by the change in slope of the median trend. When the LCBLH maxima are higher
than the JFJ, the in-situ instrumentation are reached by undiluted, aerosol-laden LCBL air and the absorption
coefficient α grows even more. In addition to the slope of the median trend, it is important to explain the
interquartile variability of each box and their physical meaning. The first box centred at about 2000 m shows a
large interquartile range of α values, this is due to Saharan dust events occurring mainly during Autumn and
winter above the LCBLH and increasing significantly the value of α. The box centred at the JFJ height also
shows a large interquartile range of α values, in this case the variability is due to the large α values
corresponding to the LCBLH higher than the JFJ and the smaller α values corresponding to the LCBLH lower
than the JFJ. In conclusion, Figure 7 shows clearly the impact of the LCBL air on the absorption coefficient α
measured at the JFJ.

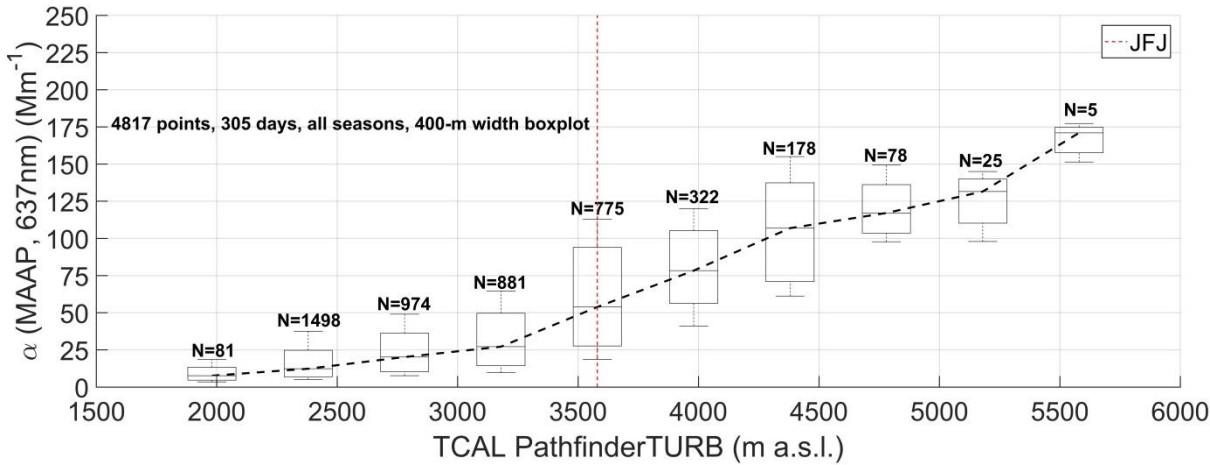

Figure 8. Box-plot showing the relation between the absorption coefficient at 637 nm measured by the MAAP at the JFJ vs the LCBLH retrieved by PathfinderTURB for the period September 2014 to November 2015.

In the same way as in Figure 7 for the LCBLH, Figure 8 shows the relation between α and the TCAL. Differently from Figure 7, not only the maxima of α and TCAL are shown in the box-plot, but all the hourly data from all seasons. The TCAL represents the upper boundary of the AL, when the TCAL is below the JFJ the in-situ instrumentation on the JFJ is located inside the FT showing very little absorption. Within the range of altitudes 2000-3380 m, the slope of the median trend is smaller than the one in Figure 7, this is because even when the TCAL grows deeper towards the JFJ, the strength of the injections coming from beneath the AL is insufficient to influence significantly the absorption measurements. As for the LCBL, also for the TCAL when it reaches the JFJ the α values become larger and the slope changes accordingly. Because the aerosols injected into the AL do not undergo a convective mixing, they tend to settle under the gravity force leading to higher aerosol concentration at the bottom than at the top of the AL. Also for this reason the absorption grows larger accordingly to a higher TCAL (3380-4580 m) and the slope of the trend remain almost constant showing the linearity of the physical process. For higher altitudes ($z > 4580$ m) α continues to grow, but at a lower rate and with a decreasing number of occurrences. Also for the TCAL- α relation the interquartile range of the box centred at the height of the JFJ is larger than the other boxes showing larger values of α for TCAL > JFJ and smaller α for TCAL < JFJ.

In order to summarizing the results shown in Figures 7 and 8, we provide in Figure 9 the overall impact of the LCBL, CAL and FT on the in-situ measurements of α. Each box collects all data from all seasons corresponding to the background atmosphere (FT), the partially mixed air (CAL) and the undiluted, aerosol-laden air (LCBL) with respect to the position of the JFJ. This box-plot represents perfectly the impact of the three atmospheric regions and confirms the importance of an automatic monitoring of the atmosphere at the JFJ and in general in the mountainous regions where the dynamics are complex due to topography and wind circulation.

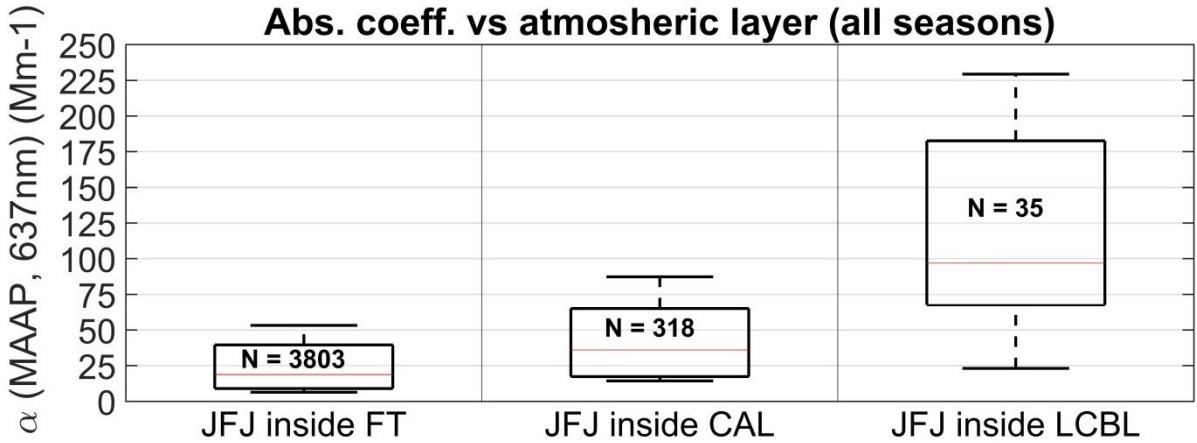

Figure 9. Box-plot of α and FT, CAL, LCBL pairs. Each portion collects all pairs over the corresponding
atmospheric region for all seasons.
**7 Conclusions**
A novel algorithm, PathfinderTURB, has been developed, validated and applied to retrieve the vertical structure
of the planetary boundary layer. PathfinderTURB provides reliable estimates of the daytime convective
boundary layer height (CBLH) and of the Top of the Continuous Aerosol Layer (TCAL) operationally and
without need of ancillary data or any a priori information (except for climatological limits) from a model.
PathfinderTURB can also be adapted to different probing line's angles and types of instrument. For this study,
two settings have been tested and applied to the data of two CHM15k, the vertical-pointing and tilted-pointing.
PathfinderTURB has been applied to one year of data measured by two CHM15k ceilometers operated at the
Aerological Observatory of Payerne, on the Swiss Plateau, and at the Kleine Scheidegg, in the Swiss Alps. The
algorithm has been thoroughly evaluated and validated at Payerne. The CBLH retrievals obtained by
PathfinderTURB have been compared against two references, (i) the manual detections by human experts and
(ii) the noon CBLH values retrieved by two methods based on radiosounding data: the parcel method and the
bulk Richardson method. Based on the excellent agreement with the two references, PathfinderTURB has been
applied to the ceilometer's backscatter profiles between the Kleine Scheidegg and the Jungfraujoch (JFJ) for the
period September 2014-November 2015. The real-time monitoring of the local CBL (LCBL) and the TCAL at
the JFJ has allowed to quantify the occurrence of these two layers and to understand their impact on the
absorption coefficient, α, measured in-situ at JFJ. The results have shown that the CAL reaches or includes the
JFJ during the 40.92% of the time in summer and the 21.23% in winter for a total of 97 days during the two
seasons. The LCBL reaches or includes the JFJ for short periods (3.94% of the time) on 20 days in summer and
never during winter. The impact of the LCBL and CAL on the in-situ measurements of α at the JFJ is
unambiguously shown in Figures 7 and 8 for different ranges of altitudes. The relation of the LCBLH- α and
TCAL- α pairs is linear, but with different slopes for altitudes below and above the JFJ, with a clear modification
of α due to the injections of the LCBL air into the aerosol layer (AL) reaching the in-situ instrumentation at the
JFJ. In a more general way, the overall impact of the LCBL, CAL and FT on the in-situ measurements of α is
shown in figure 9. As expected the LCBL modifies more the in-situ measurements at the JFJ in terms of absolute
value of α, but it is outnumbered by a factor of 10 in terms of occurrences by the CAL. The CAL is in fact more

diluted than the LCBL but embeds the JFJ 10 times more frequently than the LCBL and then its impact on the in-situ measurements is significant. The rest of the time the JFJ is within the FT with values of absorption characteristic of a molecular atmosphere.

The results obtained at KSE and JFJ, are in agreement and confirm the indirect measurements and model simulations of previous works, especially those by Zellweger et al. (2003), Collaud Coen et al. (2011), Ketterer et al. (2014) and Herrmann et al. (2015). Differently from the previous, our study has provided for the first time the possibility to calculate the occurrences of the convective (LCBL) and injection (AL) layers directly at the JFJ. The added value is the real-time application of PathfinderTURB to the ceilometer profiles connecting KSE to the JFJ and the possibility to have automatic LCBLH and the TCAL values at the JFJ. Indeed, before our study, LIDARs, ceilometers and wind profilers have always been used for vertical probing at a fixed distance (5-15 km) from the JFJ, which required stringent assumptions about the homogeneity of the atmosphere between the measurement site and the JFJ. The results presented have proven the importance of an automatic monitoring of the atmosphere at the JFJ and in general in the mountainous regions where the dynamics are complex due to topography and wind circulation.

In perspective, based on the adaptability of PathfinderTURB to diverse topographic conditions and on the fact that it does not require real-time ancillary data, PathfinderTURB is best suited to treat a large dataset from networks of ceilometers in real time.

**Acknowledgements**

This study has been financially supported by the SNF through ICOS-CH. The authors would further like to thank Nicolas Bukowiecki for giving access to JFJ aerosol measurement data. The authors are grateful to Kornelia Pönitz and Holger Wille (Lufft) for technical information about the CHM15k.

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

1    **List of Acronyms**

| Acronym | Description |
| --- | --- |
| AL | Aerosol Layer |
| BL | Boundary Layer |
| bR | bulk Richardson |
| CAL | Continuous Aerosol Layer |
| CBH | Cloud Base Height |
| CBL | Convective Boundary Layer |
| CBLH | Convective Boundary Layer Height |
| EZ | Entrainment Zone |
| FT | Free Troposphere |
| JFJ | Jungfraujoch |
| KSE | Kleine Scheidegg |
| LCBL | Local Convective Boundary Layer |
| LCBLH | LCBL Height |
| LIDAR | Light Detection And Ranging |
| liminf | Minimum altitude limit |
| limphys | physically meaningful altitude limit |
| MAAP | Multi-Angle Absorption Photometer |
| ML | Mixed Layer |
| MWR | MicroWave Radiometer |
| PathfinderTURB | Pathfinder method based on turbulence |
| PAY | Payerne |
| PM | Parcel Method |
| $Ri_b$ | bulk Richardson number |
| RL | Residual Layer |
| RS | Radiosonde |
| SNR | Signal-to-Noise Ratio |
| TCAL | Top of Continuous Aerosol Layer |

