# Peer review of "PathfinderTURB: an automatic boundary layer algorithm."

_Atmospheric Chemistry and Physics, 2016_

## Referee Comment (RC1) · Anonymous Referee #2 · 11 Apr 2017

This manuscript deals with the retrieval of the convective boundary layer (CBL) height and the continuous aerosol layer (CAL) from ceilometer backscatter data using the PathfinderTURB algorithm. The result of this method were compared to manual detection of the convective boundary layer height and the results of the Richardson's method applied on Radiosonde data. After a successful validation the PathfinderTURB algorithm was applied to a tilted ceilometer in a location with extreme terrain variations in order to detect the height of the CBL and CAL above the Sphinx observatory. In this observatory, in-situ aerosol measurements are carried out which can usually be assigned to take place in the free atmosphere. Along the way, the authors utilize an impressive

data set from different remote and in-situ sensors. The data and results themselves are highly interesting.

However, the manuscript has several weaknesses and major revisions are needed. In general the core concepts of the manuscript are presented in a vague and confusing way with many repetitions. This manuscript as currently written will make readers work way too hard to understand the results. It contains too many misspellings and syntax mistakes. In general, could you please try to write shorter sentences (some of them exceed 60 words per sentence!). I suggest, that the manuscript needs a language editing by a native speaker. If the authors could clarify the analysis and structure, it would result in a much improved paper.

Title p. 1,l.4-5: check order of name and first name.

Abstract Could you please clarify the structure of the abstract in order to improve readability (e.g. introduction, experimental sites and instrumentation, algorithm, validation, results, conclusion)?

What is the meaning of TURB in "PathfinderTURB"?

Manuscript: The extensive use of abbreviations and symbols makes reading the manuscript hard. Could you please reduce the number of abbreviations and symbols used in the ms?

Check space between number and unit throughout the ms.

p.2, l.21-p.3, l.6: Description of the ceilometer belongs rather to 4.2.

p.3 l.10-13: "The difference between the ML and the CBL is in the 10 term "mixed" (and not "mixing") where the mixed layer indicates a layer in which the profiles 11 of potential temperature and humidity do not vary much in height and the particles and gases 12 are well-mixed, but are not necessarily still mixing." Please delete first part of the sentence. Why do you use italic type for "mixed layer"?

p.4, l.1 What do you mean with “atmospheric concentration observations“?

p.4, l.13 Delete “respectively“.

p.5, l.1-8 Maybe this belongs rather to chapter 3?

p.6 Suggestion: Describe the Pathfinder algorithm and the new PathfinderTURB algorithm in chapter 3. Please delete the description of the measuring sites. Describe the measuring sites first, so you can refer to this chapter without repeating this information several times in your ms.

p.7, l.20 “compared with the night”?

p.14, l.3 “Remote sensing and in-situ observations at Payerne and Kleine Scheidegg“. What about the JFJ?

p.14 Please refer only to the measuring devices used in this study.

p.15, l.9 Please be consistent using “m a.s.l.“.

p. 16 Information about the tilted zenith angle of the ceilometer was given several times (e.g. p16, 17, 38).

p. 16-17 Be consistent when introducing symbols (e.g. p.16, l. 12 vs. l.13, p. 17, l. 12 etc.)

p.17, l.28 ff. Maybe you put all Methods in a method section and all results in a separate chapter. Try to omit repetitions.

p.17, l.32 “We believe” ?

p.18, l.2 “Therefore, the PathfinderTURB algorithm”

p.18, l.10 Please write formulas in a convenient way (a = b+c).

p.18 Please re-write chapter 5.1 “Human expert CBLH retrieval” in a straight way, so that it is easy to understand. Concentrate on the most important facts. Do not use the

appendix for repetitions. What is the reason for the division of the experts in the test group and the reference?

p.18,l.9 Remove space between "cases" and full stop.

p. 20,l.10 Change "timeseries" to "time series"

p.21,l.4 Write "w.r.t" out.

p.22, l.1 What do you mean by "unphysical jumps"?

p.22, chapter 5.2 Could you reduce the number of symbols and abbreviations used in the text. It is way to hard to read the ms.

Be consistent when referring to Figures (Fig. X or Figure X).

p.25,l.14 Change "(Henne at al., 13 2004)This" to "(Henne at al., 13 2004) .This".

p.25-28 seems to be a combination of literature review methods and repetitions.

p.35 "amplitude (max/min)". Delete (max/min)

Figures Please be consistent in the use of brackets around units.

Fig.1: The quality of the printed graph is not very good. Especially in the third panel it is difficult to differentiate between TCAL and CBH and the background colours.

Fig.2: The aerial view is not very meaningful and could be deleted.

Fig.3 and 4 upper panel: Could you insert the standard deviation or measuring error in the scatter plots for both methods?

Fig. 6: I don't get the meaning of the left legend. Furthermore, it is not easy to detect the line of the 1h.running median TCAL. Please delete the doubled full stop in the caption (p.31, l.4).

Fig.8 and 9: Use smaller dots to minimize overlapping dots. Could you please improve readability of the symbols in the legend?

Literature Check order of name and first name (Baars et al. 2008, Balzani et al. 2008), placement of the year (Henne et al. 2010)

---

## Referee Comment (RC2) · Anonymous Referee #1 · 12 Apr 2017

Poltera et al. present in their paper a further developed algorithm and procedure to derive boundary layer heights using ceilometer measurements. The convective boundary layer (CBL) and the continuous aerosol layer (CAL) are the two main layers which are determined. Their retrieval was tested at two sites in Switzerland and evaluated using manual (expert) examination, as well as radio sondes and applying the Richardson method. Their improved retrieval results were then used to investigate the influence of the boundary layer on continuous in-situ measurements at the high-alpine site Jungfraujoch.

[Figure]

The work is described in a very detailed manner. Overall, the methods applied here appear to be sound and valid. However, there is no clear separation between the method, result and discussion part, which complicates the overall reading. It appears that the structure of the manuscript has not been thought through well (e.g. a few singular subsections exist that are not followed by matching subsections) and the manuscript often reads like an extensive lab report. The resulting lengthiness of the manuscript makes it difficult for the reader to grasp the main messages. In addition, the manuscript needs a thorough editorial read (if possible by a native speaker).

The comparison to the in-situ measurements is not very convincing and needs a thorough revision. After studying the manuscript, it is not really clear to the reviewer what new findings have actually been brought up to the table. Therefore, I recommend major revisions. I encourage the authors to substantially shorten and to improve the structure of their manuscript while focusing on their main findings. More detailed comments are given below (in arbitrary order).

Detailed comments

- The current manuscript is full of acronyms. I suggest to add a table with a summary of all acronyms at the end of the manuscript.

- Page 3: Before describing the retrieval algorithm I would suggest to add a paragraph / subsection on the instruments being used.

- Page 5, line 31: Why are different bin heights used for the two different sites? Do the numbers given here relate to the vertical or to the slant path?

- Page 6, line 10: This statement ("... the overlap of the ceilometer is normally sufficiently large ...") implies that there are exceptions. Please clarify.

- Page 7, line 7-8: Why are the gradients different for the early morning periods?

[Figure]

- Page 7, Eq. 1 and 2: Please use a more mathematically sound way to describe your formulas (i.e. avoid entire words like 'weights' and use Greek letters instead, also add the specific time and height dependencies).

- Page 9: Second sentence is a repetition from page 6.

- Figure 1: The first and last panel show more or less the same thing and could be combined. Avoid the sub-panel titles since the date is already given in the caption. Please define the acronyms in the legend once more in the figure caption. The colour scheme in panel c is not very suitable to detect the overlaying retrieval results.

- Page 12, line 9: Here, the full overlap is stated to be at 800 m while on page 6 it is stated to be 350 m. Please clarify.

- Equation 3: It should be $S(r,t)$ to be consistent.

- Page 13: Section 4.2.1 is not followed by a section 4.2.2 as one would expect. There are also some repetitions in that section which should be removed (e.g. the information on the tilted angle). Parts of this paragraph are of motivative nature and should be moved to the introduction.

- Figure 3: Are both plots really needed here? The lower plot could be moved to the supplement and the key-numbers could be mentioned in the text. The size of the figures is quite large and could be decreased. Please improve the figure caption by avoiding individual acronyms (like 'manualPBL') and by writing full meaningful sentences so that the reader understands the figure without going to the text and finding the acronyms.

- Page 17, line 18: $bR$ is not defined yet.
- Figure 4: Again, these two panel figures can be reduced to one main figure. The lower panel could be moved to the supplement. Please add uncertainty bars in the scatter plots. The caption is also not consistent with the axis labelling (in the figure MLH$_{bR}$ is shown which is called RS($bR$ 12H) in the caption).

- Page 20: Again, no section 6.1.2 is followed after 6.1.1. Have the authors really carefully thought about the structure of their manuscript?

- Sect 6.3.3, Table 1 and Figure 7: I have my doubts that these results are really robust and trustworthy. On page 22, line 2-5, the authors state that the results of the LCBLH retrieval for the winter months were not taken into account due to the lack of statistical significance. However, they are now (in terms of the LCBL and CAL reaching JFJ) discussed in detailed and presented in Table 1 and Figure 7. If the algorithm can't retrieve the height of the LCBL, how can you be sure that the JFJ is inside the specific layers? This part needs to be thoroughly revised by adding statistical values (like data coverage) to the table. The reader needs to know how trustworthy these values are. Figure 7 can be omitted since it is a repetition of Table 1 and nothing new is learned from it.

- Section 6.4: The comparison to the in-situ measurements is not very convincing. Why did the authors choose the absorption measurements (MAAP) instead of the scattering measurements (nephelometer)? Most of the signal of the ceilometer comes from particle scattering (backscattering) so I would assume it is more related to the nephelometer measurements. Figure 8 is also not convincing at all. All individual seasons show almost no correlation, while the improved annual correlation is therefore only a result of the overall seasonal variations. Besides the fact that the axis ranges in Fig. 8 are poorly chosen, I still don't understand why one would expect a linear relationship. The linear relationships are not at all clearly seen in the figures. Please clarify and revise.

- Figure 9: This figure is very difficult to interpret. The differences in the specific

seasons are impossible to distinguish. Why would one expect a linear relationship? Maybe plot the seasons separately. The y-axis range should be improved.

- The conclusion part has to be revised and shortened to the main findings. Currently it is just a repetition of the result section. What have we actually learned?

- In all figures with linear regressions: Please specify which regression type (orthogonal?) has been applied.

- For many of the figures it is not clear which averaging time or temporal resolution was used. Also an uncertainty analysis (error bars) are missing which should be added.

Minor comments

1. Abstract and beginning of section 4: CHM15k is not properly defined.

2. Page 9: Add the specific figure number before the panel label (i.e. Fig. 1a).

3. Page 2, line 31: Define 'TURB' at its first occurrence.

4. Page 1, line 25: Operating the ceilometer/lidar in a tilted mode is actually not so novel and has been performed at Kleine Scheidegg in previous work (see Zieger et al.(2012), Spatial variation of aerosol optical properties around the high-alpine site Jungfraujoch (3580 m a.s.l.), Atmos. Chem. Phys., 1, 7231-7249, doi:10.5194/acp-12-7231-2012).

5. Page 11, line 24: Remove KSE from the parenthesis.

6. Page 17, line 33: Mention the specific panel labels.

7. Page 24, line 3: Two dots

8. Page 24, line 15: I would call them pie charts and not circle charts.

9. Page 25. line 7: I would not use the word 'polluted' here.

10. Page 27, line 2: Add the missing figure number.

11. Sect. 6.4 and throughout the text: Define the height of TCAL as TCALH to be consistent with LCBLH.

12. Figure 9: The superscript in the y-axis label is not properly set.

13. All figures: Please be consistent with the format of your axis labels (i.e. obeying the case sensitivity). If you use acronyms, please define them once more in the figure caption.

---

## Author Response (AR1)

Dear Reviewer #1,

We thank you for the detailed revision of our manuscript. We have considered all your suggestions and we have modified the manuscript accordingly. We believe that the manuscript has been significantly improved in clarity of the text, readability and visualisation of the data in the graphics. Below we provide the answers to the general and detailed comments of the Reviewer.

**General comments**

*It appears that the structure of the manuscript has not been thought through well (e.g. a few singular subsections exist that are not followed by matching subsections) and the manuscript often reads like an extensive lab report. The resulting lengthiness of the manuscript makes it difficult for the reader to grasp the main messages. In addition, the manuscript needs a thorough editorial read (if possible by a native speaker).*

A thorough editorial revision of the manuscript has been done. The whole text has been improved for the level of English, the readability and clearness of the independent sections. Many sections have been merged and the stand-alone sub-sections have disappeared. Parts that were too technical and did not need to stay in the manuscript have been moved to the supplement.

*The comparison to the in-situ measurements is not very convincing and needs a thorough revision. After studying the manuscript, it is not really clear to the reviewer what new findings have actually been brought up to the table.*

The comparison of the retrieved layers by PathfinderTURB  with the in-situ measurements at Jungfraujoch is now easier to understand and the correlation between the two datasets is straightforward. Figures 6-9 have been completely re-done.

**Point-by-point answer to the Reviewer**

*The current manuscript is full of acronyms. I suggest to add a table with a summary of all acronyms at the end of the manuscript.*

Done.

*Page 3: Before describing the retrieval algorithm I would suggest to add a paragraph/ subsection on the instruments being used.*

We agree with the Reviewer that a description of the used ceilometer should go before the section describing the PathfinderTURB,  we have then moved section 4 after section 2 and right before section 3. Section 4 (now Section 3) has been re-written to improve the readability and avoid all repetitions that occurred in the previous version. The overall section has also been shortened.

*Page 5, line 31: Why are different bin heights used for the two different sites? Do the numbers given here relate to the vertical or to the slant path?*

We added a phrase to the text to explain the different resolutions used. The different range and time resolutions, with longer integration at KSE, are necessary to compensate the slant path effect and the fact that at KSE the SNR is normally smaller than at PAY due to the smaller concentration of aerosols at these high altitude.

*Page 6, line 10: This statement ("... the overlap of the ceilometer is normally sufficiently large ...") implies that there are exceptions. Please clarify.*

Yes, each and every ceilometer can have a different receiver-transmitter overlap function, the bi-static ceilometer like the CHM15k can have as large overlap as 0.5 at 350 m, which means that physical measurements of range corrected signal can be performed already at that level.

*Page 7, line 7-8: Why are the gradients different for the early morning periods?*

The text has been modified and improved for clarity. A full explanation about why we use different thresholds has also been added.

*Page 7, Eq. 1 and 2: Please use a more mathematically sound way to describe your formulas (i.e. avoid entire words like 'weights' and use Greek letters instead, also add the specific time and height dependencies).*

Done

*Page 9: Second sentence is a repetition from page 6.*

We have modified the text.

*Figure 1: The first and last panel show more or less the same thing and could be combined.*

We have preferred to keep the two panels separated in order not to start directly with a panel with many parameters displayed and the weights.

*Avoid the sub-panel titles since the date is already given in the caption.*

Done

*Please define the acronyms in the legend once more in the figure caption. The colour scheme in panel c is not very suitable to detect the overlaying retrieval results.*

They all are defined in the current and previous sections.

*Page 12, line 9: Here, the full overlap is stated to be at 800 m while on page 6 it is stated to be 350 m. Please clarify.*

Here we speak about the full overlap, which for the KSE and PAY ceilometers corresponds to 800 m. Before in the text, the 350 m is just a level situated well above the first overlap point (80 m) and at which physical measurements of range corrected signal can be performed.

*Equation 3: It should be S(r; t) to be consistent.*

Done

*Page 13: Section 4.2.1 is not followed by a section 4.2.2 as one would expect. There are also some repetitions in that section which should be removed (e.g. the information on the tilted angle). Parts of this paragraph are of motivative nature and should be moved to the introduction.*

We have moved section 4 after section 2 and right before section 3. Section 4 (now Section 3) has been re-written to improve the readability and avoid all repetitions that occurred in the previous version.

*Figure 3: Are both plots really needed here? The lower plot could be moved to the supplement and the key-numbers could be mentioned in the text. The size of the figures is quite large and could be decreased. Please improve the figure caption by avoiding individual acronyms (like 'manualPBL') and by writing full meaningful sentences so that the reader understands the figure without going to the text and finding the acronyms.*

We believe that the two plots bring different point of views that allow the reader to understand the different density regions of the comparison between manual and automatic retrieval of the CBLH (top panel), and to have a more detailed and quantitative information about the Gaussian-like shape of the distribution, all statistics and the distribution of the outliers (bottom panel). We have then decided to keep both panels. The caption has been improved for readability.

*Page 17, line 18: bR is not defined yet.*

The *bR* method is now defined in Section 6.1.

*Figure 4: Again, these two panel figures can be reduced to one main figure. The lower panel could be moved to the supplement.*

Same as above

*Please add uncertainty bars in the scatter plots. The caption is also not consistent with the axis labelling (in the figure MLHbR is shown which is called RS(bR 12H) in the caption).*

Done

*Page 20: Again, no section 6.1.2 is followed after 6.1.1. Have the authors really carefully thought about the structure of their manuscript?*

We removed the sub-section title.

*Sect 6.3.3, Table 1 and Figure 7: I have my doubts that these results are really robust and trustworthy. On page 22, line 2-5, the authors state that the results of the LCBLH retrieval for the winter months were not taken into account due to the lack of statistical significance. However, they are now (in terms of the LCBL and CAL reaching JFJ) discussed in detailed and presented in Table 1 and Figure 7.*

Figure 7 has been removed (as requested also by the other Reviewer). Figure 6 shows now also the winter daily cycle of TCAL and LCBLH.

*If the algorithm can't retrieve the height of the LCBL, how can you be sure that the JFJ is inside the specific layers? This part needs to be thoroughly revised by adding statistical values (like data coverage) to the table. The reader needs to know how trustworthy these values are.*

The algorithm PathfinderTURB have no problem retrieving the LCBL when the layer exists and the ceilometer shows it. The winter statistic consist of less data just because the LCBL can rise above the

KSE only rarely during winter. Nevertheless, we have decided to show the winter average daily cycle in Figure 6d and to discuss quantitatively the related statistics. The entire Section 6. has been rewritten accordingly to the new figures and to improve the text clarity.

*Figure 7 can be omitted since it is a repetition of Table 1 and nothing new is learned from it.*

Figure 7 has been removed.

*Section 6.4: The comparison to the in-situ measurements is not very convincing. Why did the authors choose the absorption measurements (MAAP) instead of the scattering measurements (nephelometer)? Most of the signal of the ceilometer comes from particle scattering (backscattering) so I would assume it is more related to the nephelometer measurements.*

The author's intention is to study the impact of an aerosols layer on an in-situ measurement that provide a proxy for the presence of the layer itself. In this sense, any extensive aerosol property is an excellent proxy for the presence of an aerosol layer. Moreover, the absorption coefficient is indirectly a measurement of the black carbon concentration, which is a good proxy for CBL air. Incidentally, for most of the studied period, the nephelometer measurements were not available.

*Figure 8 is also not convincing at all. All individual seasons show almost no correlation, while the improved annual correlation is therefore only a result of the overall seasonal variations. Besides the fact that the axis ranges in Fig. 8 are poorly chosen, I still don't understand why one would expect a linear relationship. The linear relationships are not at all clearly seen in the figures. Please clarify and revise.*

We agree with the Reviewer that the previous Figure 8 was difficult to read and did not bring a straightforward message about the relation between the TCAL/LCBLH and the in-situ measurements. We have decided to change completely the visualization of the sought relation and we have found a clear and robust way to show this relation. The majority of the text has been changed accordingly to the changed figures. The current figures 7, 8 and 9 have now the same layout and are very consistent with each other.

*Figure 9: This figure is very difficult to interpret. The differences in the specific seasons are impossible to distinguish. Why would one expect a linear relationship? Maybe plot the seasons separately. The y-axis range should be improved.*

As explained above

*The conclusion part has to be revised and shortened to the main findings. Currently it is just a repetition of the result section. What have we actually learned?*

The Conclusions have been entirely re-written.

*In all figures with linear regressions: Please specify which regression type (orthogonal?) has been applied.*

No linear regression has been performed anymore in the new plots. So no confusion will arise from it.

*For many of the figures it is not clear which averaging time or temporal resolution was used. Also an uncertainty analysis (error bars) are missing which should be added.*

All necessary information about data clustering, temporal resolution and uncertainty is now provided for Figures 7, 8 and 9.

**Minor comments**

*Abstract and beginning of section 4: CHM15k is not properly defined.*

It is now specified that is a specific type of ceilometer.

*Page 9: Add the specific figure number before the panel label (i.e. Fig. 1a).*

Done

*Page 2, line 31: Define 'TURB' at its first occurrence.*

Done

*Page 1, line 25: Operating the ceilometer/lidar in a tilted mode is actually not so novel and has been performed at Kleine Scheidegg in previous work (see Zieger et al.(2012), Spatial variation of aerosol optical properties around the highalpine site Jungfraujoch (3580 m a.s.l.), Atmos. Chem. Phys., 1, 7231-7249, doi:10.5194/acp-12-7231-2012).*

Thanks to the Reviewer for highlighting the missing reference. Zieger et al have already used a tilted LIDAR at JFJ but only for 9 days back in 2010. What was missing then was the creation of a dataset large enough to create a statistics. The study is now cited and discussed in the introduction.

*Page 11, line 24: Remove KSE from the parenthesis.*

Done

*Page 17, line 33: Mention the specific panel labels.*

Done

*Page 24, line 3: Two dots*

All figures have now a full stop punctuation in the caption (Figure XX.)

*Page 24, line 15: I would call them pie charts and not circle charts.*

Figure 7 has been removed

*Page 25. line 7: I would not use the word 'polluted' here.*

"polluted" has been replaced by "aerosol-laden"

*Page 27, line 2: Add the missing figure number.*

Done

*Sect. 6.4 and throughout the text: Define the height of TCAL as TCALH to be consistent with LCBLH.*

TCAL is already a height, it is the Top (height) of the CAL.

*Figure 9: The superscript in the y-axis label is not properly set.*

Figure 9 has been replaced with a new one.

*All figures: Please be consistent with the format of your axis labels (i.e. obeying the case sensitivity). If you use acronyms, please define them once more in the figure caption.*

Done.

Dear Reviewer #2,

We thank you for the detailed revision of our manuscript. We have considered all your suggestions and we have modified the manuscript accordingly. We have noticed that your review was based on the version of the paper before the technical revision. Nevertheless, the scientific content and the structure of the paper have not changed much after the technical revision. We have then proceeded to the revision of the manuscript based on your comments.

Below we provide the answers to the general and detailed comments of the Reviewer.

**General comments**

*In general the core concepts of the manuscript are presented in a vague and confusing way with many repetitions. This manuscript as currently written will make readers work way too hard to understand the results. It contains too many misspellings and syntax mistakes. In general, could you please try to write shorter sentences (some of them exceed 60 words per sentence!). I suggest, that the manuscript needs a language editing by a native speaker. If the authors could clarify the analysis and structure, it would result in a much improved paper.*

A thorough editorial revision of the manuscript has been done. The whole text has been improved for the written English, the readability and clearness of the independent sections. Many sections have been merged in order to clarify the manuscript's structure and all repetitions have been removed. Parts that were too technical and did not need to stay in the manuscript have been moved to the supplement.

**Detailed comments**

*Title p. 1,l.4-5: check order of name and first name.*

Done.

*Abstract Could you please clarify the structure of the abstract in order to improve readability (e.g. introduction, experimental sites and instrumentation, algorithm, validation, results, conclusion)?*

The abstract has been rewritten and shortened. It now reads in a much clearer way.

*What is the meaning of TURB in "PathfinderTURB"?*

The meaning of the name PathfinderTURB is now provided directly in the introduction.

*Manuscript: The extensive use of abbreviations and symbols makes reading the manuscript hard. Could you please reduce the number of abbreviations and symbols used in the ms?*

We added a list of acronyms at the end of the manuscript to help finding the meaning of all of them in a same place. Moreover, we tried to limit the use of acronyms where possible throughout the text.

*Check space between number and unit throughout the ms.*

Done.

*Check space between number and unit throughout the ms.*

We have moved section 4 after section 2 and right before section 3. Section 4 (now Section 3) has been re-written to improve the readability and avoid all repetitions that occurred in the previous version.

*p.3 l.10-13: "The difference between the ML and the CBL is in the 10 term "mixed" (and not "mixing") where the mixed layer indicates a layer in which the profiles 11 of potential temperature and humidity do not vary much in height and the particles and gases 12 are well-mixed, but are not necessarily still mixing." Please delete first part of the sentence. Why do you use italic type for "mixed layer"?.*

The sentence has been removed.

*p.4, l.1 What do you mean with "atmospheric concentration observations"?*

The sentence has been replaced by: "observations of atmospheric compounds at different concentrations".

*p.4, l.13 Delete "respectively".*

Done

*p.5, l.1-8 Maybe this belongs rather to chapter 3?*

The paragraph has been shortened and rephrased.

*p.6 Suggestion: Describe the Pathfinder algorithm and the new PathfinderTURB algorithm in chapter 3. Please delete the description of the measuring sites. Describe the measuring sites first, so you can refer to this chapter without repeating this information several times in your ms.*

Done.

*p.7, l.20 "compared with the night"?*

The phrase has been replaced by "…the SNR drops below the value 0.6745 already at low altitudes due to the enhanced solar background".

*p.14, l.3 "Remote sensing and in-situ observations at Payerne and Kleine Scheidegg". What about the JFJ?*

The title of the new Section 3 is now: 3 Description of instruments and sites.

*p.14 Please refer only to the measuring devices used in this study.*

Done.

*p.15, l.9 Please be consistent using "m a.s.l.".*

Done.

*p. 16 Information about the tilted zenith angle of the ceilometer was given several times (e.g. p16, 17, 38).*

Repetitions have been removed.

*p. 16-17 Be consistent when introducing symbols (e.g. p.16, l. 12 vs. l.13, p. 17, l. 12 etc.).*

Symbols in the equations and in the text are now consistent with each other.

*p.17, l.28 ff. Maybe you put all Methods in a method section and all results in a separate chapter. Try to omit repetitions*

Section 5 has been re-written and parts that were too technical and did not bring essential information to the reader have been moved to the Supplement . We tried to remove all repetitions.

*p.17, l.32 "We believe" ?.*

We have removed it.

*p.18, l.2 "Therefore, the PathfinderTURB algorithm".*

Done.

*p.18, l.10 Please write formulas in a convenient way (a = b+c).*

The equation has been removed.

*p.18 Please re-write chapter 5.1 "Human expert CBLH retrieval" in a straight way, so that it is easy to understand. Concentrate on the most important facts. Do not use the appendix for repetitions. What is the reason for the division of the experts in the test group and the reference?*

Section 5 has been re-written and parts that were too technical and did not bring essential information to the reader have been moved to the Supplement . We tried to remove all repetitions.

*p.18,l.9 Remove space between "cases" and full stop.*

Done

*p. 20,l.10 Change "timeseries" to "time series".*

Done.

*p.21,l.4 Write "w.r.t" out.*

Done

*p.22, l.1 What do you mean by "unphysical jumps"?*

*the sentence has been removed*

*p.22, chapter 5.2 Could you reduce the number of symbols and abbreviations used in the text. It is way too hard to read the ms.*

We added a list of acronyms at the end of the manuscript to help finding the meaning of all of them in a same place. Moreover, we tried to limit the use of acronyms where possible throughout the text.

*Be consistent when referring to Figures (Fig. X or Figure X).*

Done

*p.25,l.14 Change "(Henne at al., 13 2004)This" to "(Henne at al., 13 2004) .This".*

Done

*p.25-28 seems to be a combination of literature review methods and repetitions.*

We have partially re-written and shortened Section 6. Nevertheless, the first part of Section 6 before section 6.1 it's fundamental to explain the dynamics occurring in complex topography.

*p.35 "amplitude (max/min)". Delete (max/min)*

The whole section does not exist anymore, it has been entirely rewritten.

*Fig.1: The quality of the printed graph is not very good. Especially in the third panel it is difficult to differentiate between TCAL and CBH and the background colours.*

Figure 1 (now Figure 2) has been edited and the quality improved.

*Fig.2: The aerial view is not very meaningful and could be deleted.*

Done.

*Fig.3 and 4 upper panel: Could you insert the standard deviation or measuring error in*

*the scatter plots for both methods?*

Done.

*Fig.3 and 4 upper panel: Could you insert the standard deviation or measuring error in*

*the scatter plots for both methods?*

We added error bars in Figure 4a. The error bars were not possible in Figure 3a, but the RMSE is indicated.

*Fig. 6: I don't get the meaning of the left legend. Furthermore, it is not easy to detect the line of the 1h.running median TCAL. Please delete the doubled full stop in the caption (p.31, l.4).*

The "iqr" stays for inter-quartile range, this is now clearly stated in the text and the caption. We have enhanced the DPI of the figure and the 1h-running mean line is now more visible. We removed the double full stop.

*Fig.8 and 9: Use smaller dots to minimize overlapping dots. Could you please improve readability of the symbols in the legend?*

Figures 8 and 9 have been replaced by new figures.

*Literature Check order of name and first name (Baars et al. 2008, Balzani et al. 2008), placement of the year (Henne et al. 2010)*

Done

[revised manuscript text omitted]

height (MLH) retrieval. In order to respond to these requirementsaddress the attribution problem, we have further developed, validated and applied the *pathfinder* algorithm originally described by de Bruine et al. (2016). We have then validated our own-developed version of pathfinder algorithm and optimized it for theapplied to real-time detectiondetections of the vertical structure of the CBL. The newBL above complex terrain. This improved version of the pathfinder, algorithm, is called PathfinderTURB, is a gradient (pathfinder algorithm based layer detection algorithm that, in addition to the traditional gradient detection, makes on TURBulence), to highlight the use of the aerosol distribution temporal variability (variance) to detect the MLH. The validatedBL height. PathfinderTURB has been applied to the data of a ceilometer installed at the Kleine Scheidegg to probe the air that is sampled by the in-situ instrumentation at the high Alpine station Jungfraujoch (JFJ). The JFJ is part of numerous global observation programs like GAW (Global Atmospheric Watch), EMEP (European Monitoring and Evaluation Programme), NDACC (Network for the Detection of Atmospheric Composition Change) and AGAGE (Advanced Global Atmospheric Gases Experiment). Most importantly, in the context of this study, JFJ participates with in-situ observations as a level-1 station in the ICOS project. In contrast to other ICOS sites located over flat terrain, it was decided to install the ceilometer at KSE to characterize the CBL below and above the JFJ. The presence of the aerosols detected by the ceilometer and the frequency at which these reach the JFJ, are directly compared to the optical, chemical and physical in-situ measurements of aerosols and trace gases at the JFJ. Several in-situ instruments are installed at the JFJ and operate continuously since many years to measure aerosols, trace gases and several meteorological parameters (Bukowiecki et al., 2016). Instruments of direct interest to our study are: a condensation particle counter (CPC; TSI Inc., Model 3772), which measures the particle number concentration and two instruments providing aerosol absorption coefficients: a Multi-Angle Absorption Photometer (MAAP) measuring at 637 nm and an aethalometer (AE-31, Magee Scientific) measuring at seven different wavelengths. The fact of measuring remotely with a ceilometer the presence of the ML directly at the JFJ andCBL air in real time provides an and close to the JFJ for more than one year is unprecedented possibility. A recent study by Zieger et al (2012) has used a scanning LIDAR tilted at 60° Zenith angle for 9 days to probe the air close to the JFJ . Also based on the results of the study by Zieger, we have decided to improve their instrument set-up and to install a ceilometer probing even closer (few meters) to the JFJ and for more than one year. This has allowed to create a statistics of CBL-events and to describe quantitatively the relation between the MLCBL dynamics and the aerosols measured in situ at JFJ. An additional motivation comes from the fact that the JFJ contributes with in situ observations of greenhouse gases as a level-1 station to the ICOS (Integrated Carbon Observation System) infrastructure. At level-1 ICOS stations (rising and falling) and the aerosols optical properties at JFJ. The relevance of such measurements becomes also clear in the framework of ICOS, where the detection of the CBL height in theirthe vicinity of a level-1 ICOS stations is requireda requirement to serve as a validation dataset forvalidate the atmospheric transport models. This is crucial when atmospheric concentration observations shouldof atmospheric compounds at different concentrations must be translated into greenhouse gas fluxes between the atmosphere and the land surface.

In Sect. 2, we present an overview of existing algorithms for the retrieval of the MLH from LIDAR backscatter profiles. The novel PathfinderTURB algorithm is then detailed in Sect. 3. In Sect. 4, we describe the configuration of the sites of Payerne (Swiss plateau) and Kleine Scheidegg (Swiss Alps) where ceilometer

**2 Overview of existing algorithms**

[revised manuscript text omitted]

3     ~~In order to retrieve the MLH operationally while minimizing the uncertainty due to the attribution problem and to assure the adaptability of the algorithm to diverse topographical conditions, we have extended the pathfinder algorithm proposed by de Bruine et al. (2016) by adding a variance criterion and the retrieval of the Continuous Aerosol Layer (CAL). The extended version 
[revised manuscript text omitted]

9 4    Remote sensing and in-situ observations at Payerne and Kleine Scheidegg

10 Two CHM15k ceilometers were installed at two different sites in Switzerland. The first at PAY in a rural and

11 comparatively flat environment. PAY is equipped with numerous meteorological measurements allowing to

12 interpret and to validate the measurements from the ceilometer. The most relevant measurements in the

13 framework of the presented study are: a Raman LIDAR measuring humidity, temperature and backscatter at 355

14 nm and at the Raman shifted wavelengths (Dinoev et al., 2013; Brocard et al., 2013), a wind profiler (Degreane,

15 1290 MHz), a RPG HATPRO microwave radiometer measuring temperature and humidity using 14 channels

16 around the water vapour and oxygen absorption lines, the operational Meteolabor SRS-C34 radiosondes

17 launched twice daily at 00 and 12 UTC (Philipona et al, 2013), the sun photometers (direct, diffuse and global

18 short and long wave, Vuilleumier et al., 2014) and the surface sensors of temperature and humidity.

19 A second instrument was installed at KSE, close to the JFJ. The JFJ is part of numerous global observation

20 programs like GAW (Global Atmospheric Watch), EMEP (European Monitoring and Evaluation Programme),

21 NDACC (Network for the Detection of Atmospheric Composition Change) and AGAGE (Advanced Global

Atmospheric Gases Experiment). Most importantly, in the context of this this work, JFJ participates with in-situ observations as a level-1 station in the ICOS project. In contrast to other ICOS sites located over flat terrain, it was decided to install the ceilometer at KSE to characterize the CBL below and above the JFJ. The presence of aerosols, detected by the ceilometer, and the frequency at which these reach the JFJ, can be directly compared with the chemical and physical in-situ measurements of aerosols and trace gases at the JFJ. In fact, several in-situ instruments are installed at the JFJ and operate continuously since many years to measure optical and chemical properties of aerosols and trace gases as well as diverse meteorological parameters (Bukowiecki et al., 2016). Instruments of direct interest to our study are: a condensation particle counter (CPC; TSI Inc., Model 3772), which measures the particle number concentration and two instruments providing aerosol absorption coefficients: a Multi-Angle Absorption Photometer (MAAP) measuring at 637 nm and an aethalometer (AE 31, Magee Scientific) measuring at seven different wavelengths. The aerosol in situ measurements are performed under dry conditions (relative humidity <20 %), while the ceilometer probe the unmodified atmospheric volume.

**4.11.1 Sites description**

The PAY site (Figure 2) is situated in the centre of the Swiss Plateau between the Jura Mountain range to the north-west (at a distance of 25 km) and the Alpine foothills to the south-east (20 km). The measurement site is characterized by a rural environment leading to biogenic aerosols sources combined with moderate urban emissions characterized by anthropogenic aerosol sources especially related to cars exhaust and house heating.

The KSE (Figure 2, KSE) is located in the Bernese Oberland Alpine region, at an altitude of 2069 m. It can be seen as a saddle point between the mountain peak Lauberhorn (2472 m) to the northwest and the Jungfraujoch (3465 m) to the southeast, and it is a pass between the villages of Wengen and Grindelwald. This topographic configuration has a considerable influence on the local wind circulation. Winds at the KSE are mostly blowing along the SW-NE axis (Ketterer et al., 2014), whereas the prevailing wind at JFJ are from NW toward SE. The JFJ itself is located on the ridge formed between the Mönch and the Jungfrau mountains and is 4.5 km to the south-east and 1.5 km higher than KSE. Most of the atmospheric observations at the JFJ are obtained at the Sphinx observatory (3580 m a.s.l.).

[Figure]

1 Figure 2: Aerial view and topography of PAY (elevation profile along the 127.2° Azimuth), and KSE (elevation
2 profile along the 151.6° Azimuth) as provided by the federal office of topography (http://www.geo.admin.ch/).
3 The red stars mark the position of the ceilometers at PAY and KSE, the black diamond marks the JFJ position.

4 **4.2 CHM15k Ceilometer data and settings**

5 The measurements from a ceilometer of type CHM15k-Nimbus (hereafter referred to as CHM15k) manufactured
6 by Lufft have been used for this study. The CHM15k is a bistatic LIDAR with a Nd:YAG solid state laser
7 emitting linearly polarized light at a wavelength of 1064 nm. It has a repetition rate ranging between 5 and 7
8 kHz, a maximum vertical resolution of 5 m, a maximum range of 15 km, a first overlap bin at 80 m and a full
9 overlap reached at 800 m (specific for KSE, and PAY ceilometers, Hervo et al., 2016). The standard instrument
10 output is a function of the received power per laser pulse, $P$, backscattered by the atmosphere at range $r$ and time
11 $t$. More precisely, the standard output of the CHM15k is the background, range and overlap corrected,
12 normalized signal, $S$ defined as:

$$S = \frac{(P(r,t) - B(t))r^2}{C_{CHM}(t)O_{CHM}(r)} \quad\quad (3)$$

14 Where, $B$ is the background, $C_{CHM}$ is a normalization factor accounting for variations in the sensitivity of the
15 receiver and $O_{CHM}$ is the temporally constant overlap function provided by the manufacturer.

The measurements used for this study have been collected during the period January-December 2014 at PAY and during September 2014 - November 2015 at KSE. At PAY, the ceilometer was setup vertically with a slight tilt (5° zenith angle) to avoid the specular reflection effect on cirrus ice crystals, as suggested by the manufacturer. At KSE, the ceilometer was mainly operated in tilted setup (71° zenith angle) in order to point towards the JFJ. At both sites, the overlap function $O_{CHM}$ has been corrected for temperature variations following (Hervo et al., 2016).

4.2.11.1 Special instrument settings for KSE

The CHM15k ceilometer at KSE was installed in August 2014 on the roof of the maintenance centre of the train station, at an altitude of 2069 m (Figure 2). From September to November 2014 and from March to November 2015, the ceilometer was tilted at 71°zenith angle with the laser beam passing close above (~20 m) the JFJ. From the beginning of November 2014 till the end of February 2015, the ceilometer was set back in the vertical position (5° zenith angle) to prevent the sun shining directly onto the ceilometer's telescope.

The tilted setup of the ceilometer was chosen to observe the injections of CBL air at the level of JFJ and to probe the same air that is measured by the in situ instruments at the JFJ. The tilted configuration has required an adjustment of the heater to redirect the heat flow inside the CHM15k case and prevent the overheating of the laser. Moreover, when measuring in slant path the maximum vertical height, $R_{max}$, depends on the tilt angle and on the instrument's maximum range (15 km for the CHM15k), then, when measuring at 19°elevation angle $R_{max} = 2.069 + 15\sin(19°) = 6.64$ km a.s.l.. At this relatively low altitude, the standard background correction that subtracts from the received power $P$ its own mean value over the *far range* (furthest ranges along the profile, where the signal is assumed to be entirely represented by noise), cannot be performed as the far range may still contain aerosols or clouds. In order to overcome this problem a new technique of background removal that depends on the variance of $P$ has been developed and applied to each profile. The variance is calculated within spatial windows of 120 m to 1600 m (in steps of 120 m) and computed for all range bins between 390 m and 14970 m. The background corresponds to the mean value of $P$ over an optimal window. The optimal window's position is the one minimizing the average of its variance values. The optimal window's width is the one corresponding to the 75[th] percentile of the variance values at the optimal window's position. Another advantage of measuring in slant path is that, compared to the vertical setup, the JFJ is reached by the ceilometer's laser beam at 4.8 km that is already in the full overlap region.

**5    PathfinderTURB validation at Payerne**

[revised manuscript text omitted]

- No precipitation (station measurement) and no fog (ceilometer measurement) for more than 2 consecutive hours.
- Only time periods of at least 30-minute duration, containing interpolated data from at least 2 experts and with spread (i.e. difference between the maximum and minimum values) of less than 10% of the mean CBLH plus 100 m are taken into account. The allowed spread increases with altitude because the probability to lie on different layers decreases with increasing altitude. The offset of 100 m added to the maximum allowed spread is an empirical value that translates into a permitted 300-m spread for a CBLH at 2000 m, which is a conservative estimate of the depth of the entrainment zone at that altitude (EZ depth can be 40% of the CBLH, Stull, 1988).

The CBLH detections respecting the above criteria are retained for comparison with PathfinderTURB. The mean value of all the valid detections (*reference* and *test group*) for each time step was calculated along with the lower and upper error bounds determined as the minimum and the maximum CBLH detections. In total, an expert consensus was reached for 135 days, out of the initial 174, covering a total of 43914 minutes. On 6 days, no ceilometer data were available. On 13 days, poor weather conditions prevented all detections. On 20 days, either the spread was too large or the duration of the matched temporal interval was too short.

Because the number of CBLH detections decreases as the days become shorter in winter, midday is the period of the day with the highest availability of detections (and the best agreement). Both morning and afternoon periods present difficulties, when detecting the CBLH from the timeseries of *S*, nevertheless the morning provides better availability of detection than the afternoon. The limitations of morning detections are related to the fact that during the first 2-3 hours after sunrise the CBL top is still low above the ground (< 200 m a.g.l.) and the overlap-corrected, normalized signal, *S*, is affected by the incomplete overlap in that zone, which makes the detection in the first few hundred meters difficult and more uncertain. Another source of complexity in the morning detections is that the residual nocturnal layer may remain aloft and very close to the developing CBLH, which increases the uncertainty related to the layer attribution. During the late afternoon, the CBL transforms into the RL as soon as buoyancy weakens and it becomes neutrally stratified. Under these conditions, the CBL top drops rapidly, but usually without displaying a clear aerosol gradient, which leads to a large (and somewhat unphysical) spread among the experts' detections.

**5.1.35.1.2    PathfinderTURB validation against the expert consensus**

After applying the ratio quality check (sect. 34.2.6) to the PathfinderTURB retrievals, the total number of the accepted retrievals covers 34720 minutes out of the 43914 minutes of obtained by the manual datadetections, i.e. 79% of the human expert consensus. In other words,The ratio quality check of PathfinderTURB removes about the 20% of itsthe retrievals because of weak gradients at the level of the retrieved CBLH. The validated PathfinderTURB retrievals that passed the quality check are spreaddistributed over the same number of days, (i.e., 135). Figure 3 shows, for Payerne and) during the year 2014.; the top panel of Figure 3 shows the density scatter plot of the CBLH values obtained at PAY by the human experts' detections meeting the (consensus and

by ) manual detections versus PathfinderTURB (top panel), and the . The boxplot plusalong with the histogram
(shown in the bottom panel) of display the differences between the two data sets. A coefficient of determination
of 0.96, an RMSE of 76 m and an interquartile range of the differences of 96 m are obtained. The median and
mean differencedifferences are 27 m and 41 m, respectively. The overestimation is largest during the second half
of the afternoon (not explicitly shown here), when PathfinderTURB tends to follow the top of the residual layer
instead of the decaying CBL. Furthermore, the error is smaller than 500 m for 98.6% of the PathfinderTURB
retrievals, and 92% of the retrievals have a relative error (w.r.twith respect to the manually determinedmanual
CBLH) smaller than 10%.

[Figure]

[Figure]

2    Figure 3 top panel shows the density. Density scatter plot of CBLH$_{PathfinderTURB}$ vs.

[revised manuscript text omitted]

**6.2    Adaptation of PathfinderTURB**

 has been adapted to use the CHM15k data along the slant-probing direction connecting KSE with JFJ. The adapted PathfinderTURB  version does not use the $VAR(S)$ profiles to calculate the weights (eq. (3)), but solely to retrieve the first transition to the enhanced turbulence zone (*liminf*), see Sect. S2 in the Supplement). In fact, at ~~PAY, PathfinderTURB does not use the signal variance when calculating the weights to determine the geodesic path at KSE, depends entirely on the tilted configuration of the ceilometer. Altitudes higher than 1000 m a.g.l. correspond already to ranges farther than 3000 m along the laser's line of sight. At KSE, the backscatter signal at these ranges is often characterized by very little aerosol concentration, and is contaminated by significant contribution from the solar background. The backscatter signal is then dominated by the noise with low SNR and the calculated variance is not reliable. At closerturbulenceto besignalthe variance~~ $VAR(S)$ can be measured reliably. At KSE, the LCBL height (LCBLH) retrieved by PathfinderTURB, corresponds to the first discontinuity in the vertical mixing of aerosols and can be estimated also during nighttime.

**6.3 Retrieval of aerosol layers at KSE **and JFJ**

The CHM15k detects the  aerosols that form in the surrounding lower-altitude valleys (e.g., 1034 m a.s.l. Grindelwald, 566 m a.s.l. Interlaken) and that are transported above the KSE. Local generation of  aerosols occurs only sparingly due to the reduced vegetation and the long periods of snow and ice cover. Nevertheless, when local aerosols production occurs, these can be transported through the ceilometer's field of view and eventually be transported up to the JFJ. The local aerosols generation and the advection from the  surrounding valleys lead to different scenarios. During daytime, both TCAL and LCBLH can be detected , the LCBLH only during periods when  the LCBL air is lifted into the

ceilometer's field of view; during by convection. During nighttime, when there is no convection, only the TCAL can be detected (if it is present). Likewise for any aerosol layers observed by the ceilometer above KSE, also the observedThe nocturnal TCAL can formstem from the residual layer offormed above the surrounding valleys (at least partially). Because. PathfinderTURB is based on the same retrieval principle during day and night, i.e.and so it looks for the first discontinuity in the well-mixeduninterrupted aerosol region,. For this reason and for simplicity we will refer to the retrieved nighttimenocturnal boundary layer also as to *mixing layer or* LCBL even when the mixing is not due to convection, but rather to mechanical mixing from the surface and katabatic winds.

6.3.16.1.1      LCBLH retrieval

The seasonal-averaged daily cycles of the retrieved LCBLH and TCAL during spring, summer and, autumn and winter are shown in Figure 6. During winter, (December 2014, January and February 2015) the algorithm has retrieved only a negligible number of LCBL heights and they have not been taken into account due to their lack of statistical significance.

During spring (Fig.Figure 6a), summer (Fig.Figure 6b) and (partially) autumn (Fig.Figure 6c), the LCBLH grows through morning till reaching a peak in the afternoon. In summer, the LCBLH has been retrieved by PathfinderTURB every day with only few exceptions. In Spring, (March-May), and in summer (June-August) the LCBL has reached the JFJ during 20 and 9 individual days, respectively. These occurrences lay above the 75 percentile of the LCBLH dataset and, hence, are not represented by the blue-shaded area in Figure 6a-b. From the systematic visual inspection and comparison of LCBLH timeseriestime series at PAY and KSE, we can say that the LCBLH peak occurs later at KSE than at the PAY. During the night, the LCBLH decreasesdrops, due to the concurrent effects of aerosol gravitational settling, subsidence and katabatic drainage flowswinds, which result from radiative cooling of the surface triggering katabatic drainage flows. A likely explanation of the delay in the onset of the LCBL and of the afternoon peak at KSE is the following:that due to the nighttime katabatic winds, driving FT air is driven down into the valley. These katabatic underneath. Depending on the season, these winds alsocan continue to blow for one or morefew hours, depending on the season, after sunrise (especially from the shaded mountain side) and work against the formation of the LCBL. The LCBLH temporal evolution follows the classical shape of a growing convective boundary layer like over flat terrain, but the growth and the duration of the LCBL occur over a shorter period. This is consistent with the delayed onset of the LCBL due to the persisting katabatic winds in the first hours of the morning and the earlier weakening of convection due to the shading effect of the surrounding mountains and the afternoon onset of the katabatic winds. This phenomenon is particularly enhanced during winter when the solar irradiance is at its minimum and the katabatic winds tend to suppress LCBL during most of the time.

Generally, theIn autumn (September-November), the LCBLH shows a less pronounced daily cycle than in spring and summer, this is probably due to the fact that the vertical transport of aerosol-rich air is reduced by the stabilization within the lower troposphere during this period (Lugauer et al., 1998). In summer, the LCBLH has been retrieved by PathfinderTURB every day with only few exceptions; from May to August the LCBLH retrievals have reached in the JFJ during short periods on 25 individual days , but these occurrences lay above the 75 percentile of the LCBLH dataset and, hence, are not represented by the blue-shaded area in Fig.6.

In winter, (December-February) PathfinderTURB could retrieve only few LCBLH because of the very stable meteorological conditions, the reduced convection and the prolonged snow and ice cover limiting the aerosol production at KSE and the surrounding valleys. For this reason the seasonal-averaged daily cycle in Figure 6d does not show any particular pattern of the LCBLH, mainly due to the very low retrieval counts.

All occurrences of when the LCBLH and TCAL have reached the JFJ during the different months are listed in the Table 1 and are shown in Figure 7 for winter and summer. Table 1. The LCBLH temporal evolution follows the classical shape of a growing convective boundary layer like over flat terrain, but the growth and the duration of the LCBL occur over a shorter period. This is consistent with the delayed onset of the LCBL due to the katabatic winds and the earlier weakening of convection under the contrasting action of the afternoon onset of the katabatic winds.

[revised manuscript text omitted]
 positive relation between LCBLH and absorption coefficient is thought to exist especially for times when the LCBL does not reach the JFJ (filled circles). In these situations, when the LCBL grows deeper and approaches the height of the JFJ, the injections of LCBL air into the AL are more likely to occur and bring more aerosols to the in-situ instruments. In this case the amplitude of the daily cycle of the absorption coefficient is greatest, as the difference between the convective peak of the LCBL (large injections) and its minimum (negligible injections) correspond to high and low absorption coefficients, respectively. On the contrary, when the LCBLH is higher than the JFJ for a few hours during the day, the residual AL is also richer in particles and the amplitude of the daily cycle of the absorption coefficient is much smaller (the absorption remains large all day). During summer and for LCBLH < JFJ (filled circles) convection causes significant injections into the AL leading to large amplitude of the daily cycle of the absorption coefficient (larger slope in Fig.8). On the other hand, during spring probably due to the thick layer of snow accumulated during winter, the convection is too weak to create significant injections into the AL, and relatively deep LCBL do not correspond to large daily cycles in the absorption (smaller slope). Despite the low coefficients of determination of the seasonal fits, when all points are fitted together the dependency of the max-min cycle of the absorption coefficient on the LCBLH becomes clearer.~~

[revised manuscript text omitted]

~~PathfinderTURB suffers anyway some limitations related to the instrument. Due to the incomplete transmitter-receiver overlap in the first few hundred meters, and the unphysical gradients occurring in this zone, PathfinderTURB, which is partly gradient-based, is affected by a larger uncertainty in the first few hundred meters. Another limitation is that during the late afternoon, the aerosols remain suspended in the air (transition from CBL to RL) showing no detectable aerosol gradient at the top of the CBL. The only detectable gradient~~

remains normally at the same altitude as the maximum CBL height reached during the central hours of the day, and corresponds more to the RL top rather than to the decaying CBL top. In fact, any method using aerosols as tracers (e.g. LIDAR) is not best suited to detect the afternoon CBL drop, but rather the RL.

The algorithmPathfinderTURB has been applied to one year of data measured by two CHM15k ceilometers operated at the Aerological Observatory of Payerne, on the Swiss Plateau, and at the Kleine Scheidegg, in the Swiss Alps. The algorithm has been first thoroughly evaluated and validated at the Payerne station. At Payerne, the. The CBLH retrievals obtained by PathfinderTURB have been compared withagainst two references, (i) the manual detections by human experts that acted as reference for the CBLH values and withand (ii) the noon CBLH values retrieved by two methods based on the thermal structure of the atmosphere and using the radiosounding data: the parcel method and the bulk Richardson method. The comparison against the human experts reference revealed a median difference of 27 m and a RMSE of 76 m. The median difference with respect to the radiosounding reference is 53 m with a RMSE of 162 m.

Based on the excellent agreement with the two reference methods, thereferences, PathfinderTURB has been applied to the complex terrain site at the ceilometer's backscatter profiles between the Kleine Scheidegg and the Jungfraujoch (JFJ) for the period September 2014-November 2015. There, The real-time monitoring of the local CBL (LCBL) is retrieved based on the data of the CHM15k whose axis has been tilted by a zenith angle of 71° in order to probe the air volume next to the Sphinx Observatory (3580 m, a.s.l.) on the Jungfraujoch.

The results presented in Section 6 showed that the PathfinderTURB can be adapted to slant-path probing, thus providing real time and continuous LCBLH and and the TCAL data along the line of sight of the CHM15k. This at the JFJ has allowed to separatequantify the contributionoccurrence of these two layers and to understand their impact on the absorption coefficient, $\alpha$, measured in-situ at JFJ.

The season averaged daily cycle shows The results have shown that the CAL reaches or includes the JFJ forduring the 40.92% of the total time in summer and for the 21.23% of the total time in winter for a total of 97 days during the two seasons. The statistics suggest that the CAL modifies the physical and chemical properties of the air sampled at JFJ, especially during summer when the absorption coefficient at 637 nm at JFJ shows a distinct dependence on the CAL depth.

The season averaged daily cycles show that theThe LCBL reaches or includes the JFJ for short periods (3.94% of the total time) on 20 days in summer and never during winter. The statistics suggest that alsoimpact of the LCBL modifies the physical and chemical properties of the air sampled at JFJ, but exclusively during summer, as theseand CAL on the in-situ measurements refer purely to the direct contact of undilutedof $\alpha$ at the JFJ is unambiguously shown in Figures 7 and 8 for different ranges of altitudes. The relation of the LCBLH- $\alpha$ and TCAL- $\alpha$ pairs is linear, but with different slopes for altitudes below and above the JFJ, with a clear modification of $\alpha$ due to the injections of the LCBL air with the JFJ. During summer the amplitude of the daily cycle of the absorption coefficient at 637 nm reveals that the amplitude is largest when the LCBLH approaches into the aerosol layer (AL) reaching the in-situ instrumentation at the JFJ and starts levelling when the LCBLH exceeds the JFJ.

As. In a more general conclusion, we can state that the way, the overall impact of the LCBL, CAL and FT on the in-situ measurements of $\alpha$ is shown in figure 9. As expected the LCBL modifies more the in-situ measurements

at the JFJ in terms of absolute value of α, but it is outnumbered by a factor of 10 in terms of occurrences by the CAL. The CAL is in fact more diluted than the LCBL but embeds the JFJ 10 times more frequently than the LCBL and then its impact on the in-situ measurements is significant. The rest of the time the JFJ is within the FT with values of absorption characteristic of a molecular atmosphere.

The results obtained at the KSE site and JFJ, are in agreement and confirm the indirect measurements and model simulations of previous works, especially those by Zellweger et al. (2003), Collaud Coen et al. (2011), Ketterer et al. (2014) and Herrmann et al. (2015). In fact for the first time the impact of the convective (LCBL) and injection (AL) layers on the in-situ measurements at JFJ has been quantitatively calculated for a complete year thanks to the automatic retrieval of the two layers directly at the JFJ. 
[revised manuscript text omitted]

---

## Author Response (AR2)

**Response to the Editor**

Dear Editor,

Thank you very much for your prompt answer and positive review. We are very satisfied by having improved the readability of the manuscript and the quality of the graphics.

All the minor comments that you have pointed out have been addressed. You can find the revised version uploaded on the website.

All the best

Giovanni Martucci

[revised manuscript text omitted]